# Neural mechanisms of feature binding in working memory
Yang Cao[1,2,6], Fuyong Chen[3,6], Hao Wang [4 ✉], Xuchu Weng [1,2 ✉], Jan Theeuwes[5] & Benchi Wang [1,2 ✉]

As a fundamental cognitive system with limited capacity, working memory (WM) strategically binds various features together to enhance its efficiency. However, the neural mechanisms governing feature binding in WM remain unsettled. Here, we employed functional magnetic resonance imaging combined with graph-based network analysis during a WM task in which participants maintained both color and location information throughout the delay period and subsequently detected and reported changes in color-location bindings versus individual features. Our results revealed a collaborative network that operates through a central workspace encompassing the somatomotor area, insula, and prefrontal cortex, underpinning the effective processing of bindings. Within these regions, we observed increased local efficiency and stronger connections during feature binding. Notably, connections within this workspace significantly correlated with behavioral performance. Among these regions, the somatomotor area, characterized by a shorter intrinsic timescale, responded more rapidly to visual input, carrying rich temporal information with more connections, and potentially served as the starting point during binding processes. These results highlight a dedicated workspace with sufficient and valid internal connections, facilitating successful binding through collaborative regional interactions.

To form an integrated and coherent representation of the visual world, we commonly bind isolated features, enabling precise perception and recollection of objects. It is worth noting that these bindings occur not only in visual perception but also in memory. For example, when faithfully memorizing a scene (e.g., a man is teaching kids how to use chopsticks) for a short period, we need to encode not only the individual features (e.g, a man, kids, and chopsticks) but also the specific combinations of these features that constitute different situations. The influential feature integration theory (FIT), originally proposed by Treisman and her colleagues[1,2], suggests that feature binding involves integrating feature maps into a shared location map[3–5]. This integration relies on spatial attention, especially when information is bound into working memory (WM)[6–9]. Such binding often results in a general deterioration in memory performance when it comes to integrating features, as opposed to memorizing individual features separately[6,10]. Nevertheless, the exact mechanism by which our brains bind features into WM remains unclear.

Early studies suggest that the hippocampus, crucial for long-term memory, is implicated in feature binding within WM[11–13]. Using functional magnetic resonance imaging (fMRI), researchers found higher hippocampal activity during feature binding compared to memorizing single features. However, other studies also identified the involvement of frontal and parietal regions in feature binding. According to the study by Sala and Courtney[14], the prefrontal cortex aids in both maintaining relevant information and facilitating feature binding through interactions between dorsal and ventral pathways. Shafritz et al.[15] found the right intraparietal sulcus and right superior parietal cortex specifically important in the encoding and maintenance of feature binding, but only when the location was task-relevant[13]. Furthermore, during early visual processing before memory engagement, feature binding may already occur in the primary visual cortex. Seymour et al.[16,17] applied multivariate pattern classification to voxel activation patterns in the primary visual cortex and found that the voxels most informative about bindings were distinct from those most informative about single features.

[1]Key Laboratory of Brain, Cognition and Education Sciences, South China Normal University, Ministry of Education, Guangzhou, China. [2]Institute for Brain Research and Rehabilitation, South China Normal University, Guangzhou, China. [3]Department of Neurosurgery, University of Hong Kong, Shenzhen Hospital, Shenzhen, China. [4]State Key Laboratory of Digital Medical Engineering, Key Laboratory of Biomedical Engineering of Hainan Province, School of Biomedical Engineering, Hainan University, Sanya, China. [5]Department of Experimental and Applied Psychology, Vrije Universiteit Amsterdam, Amsterdam, The Netherlands. [6]These authors contributed equally: Yang Cao, Fuyong Chen. ✉e-mail: haowang@hainanu.edu.cn; wengxc@psych.ac.cn; wangbenchi.swift@gmail.com

In these studies, feature binding in WM has been extensively explored through change detection tasks, which evaluate memory performance by comparing an initial set of stimuli with a second set presented after a brief delay[8,18,19]. Individuals were tasked with memorizing arrays of multi-feature objects, with the specific requirement of either binding multiple features together into WM or memorizing only a single feature. Noted, however, that comparing conditions involving multi-feature binding (e.g., color-location binding) to those involving single features (e.g., memorizing color only) may not effectively pinpoint the specific brain regions involved in feature binding, as the number of to-be-memorized features was not equitably matched between conditions (Two vs. one in the feature-binding and single-feature condition, respectively). Moreover, importantly, and as mentioned earlier, previous studies have reported the involvement of multiple brain regions in feature binding, encompassing almost the entire brain. Analyzing individual regions of interest (ROIs) may not offer a comprehensive understanding of the neural activity associated with feature binding. Instead, it is crucial to examine the relationship between distant cortical regions to grasp the intricate processes involved in feature binding.

To gain a comprehensive understanding of the neural mechanism underlying feature binding in WM, the current study also employed a change detection task with color-location conjunctions as stimuli. Specifically, a large group of participants ($N = 40$, predetermined and at least twice the size of previous studies) was asked to memorize two types of information: the bindings of color and location (i.e., the binding condition), or either the color or location information (i.e., the either-memory condition). Noted that, in the either-memory condition, participants were randomly probed with either color or location information in each trial, ensuring that the number of to-be-memorized features was comparable between conditions. In addition, the neural activities corresponding to different conditions were modeled through graph-based network analysis[20–24]. This enables us to construct functional brain networks, which have been confirmed to be essential for cognitive functions such as memory maintenance and retrieval[25,26]. This underscores the importance of studying the brain as an integrated system rather than focusing on isolated regions.

## Results

Forty participants were recruited to perform a change detection task, where they were asked to determine whether the test display differed from the memory display. The experiment comprised two conditions: (1) the either-memory condition, where changes in either color or location occurred with equal probability in the "change" trials, and participants had to identify and report the corresponding changes; (2) the binding condition, where only the color was exchanged between two colored disks in the "change" trials, resulting in modifications in color-location bindings.

### Behavioral performance

Detection sensitivity (d-prime) and mean reaction times (RTs) for each condition are presented in Fig. 1B. With condition (either-memory vs. binding) and set size (three vs. six) as factors, a repeated-measures analysis of variance on d-prime revealed significant main effects for condition, $F(1, 39) = 34.9$, $p < 0.001$, $\eta_p^2 = 0.47$, and set size, $F(1, 39) = 118.62$, $p < 0.001$, $\eta_p^2 = 0.75$. Sensitivity exhibited poorer performance in the binding condition and set size six. Furthermore, there was a trend toward a two-way interaction, $F(1, 39) = 3.98$, $p = 0.05$, $\eta_p^2 = 0.09$. Planned follow-up comparisons revealed that the d-prime was significantly lower for the binding condition than for the either-memory condition, irrespective of set sizes, both $t$s > 3.22, both $p$s < 0.01. These findings suggest that additional attentional resources were required for binding information into WM, resulting in a worse performance in the binding condition (see also previous study)[9].

Similar results were observed when participants' RTs underwent the same analysis. Significant main effects were observed for condition, $F(1, 39) = 92.1$, $p < 0.001$, $\eta_p^2 = 0.7$, and set size, $F(1, 39) = 40.6$, $p < 0.001$, $\eta_p^2 = 0.51$, as well as a statistically reliable interaction, $F(1, 39) = 4.6$, $p < 0.05$, $\eta_p^2 = 0.11$. Once again, mean RTs were longer in the binding condition relative to the either-memory condition, regardless of set sizes, both $t$s > 7.94, both $p$s < 0.001.

### Blood-oxygen-level-dependent (BOLD) signals

To determine whether previous findings focusing on individual ROIs[11,14,15,18] could be replicated under fair comparison conditions between feature binding and memorizing separate features, we employed the general linear model (GLM) on preprocessed correct data to derive the blood-oxygen-level-dependent (BOLD) response corresponding to different conditions and their respective contrasts across the entire brain. As illustrated in Fig. 2A, in the either-memory condition, substantial activity (reflected by BOLD signals) was observed primarily in four brain areas, each comprising more than 1000 voxels ($p < 0.001$, cluster family-wise error (FWE) correction). These regions were prominently located in the prefrontal cortex,

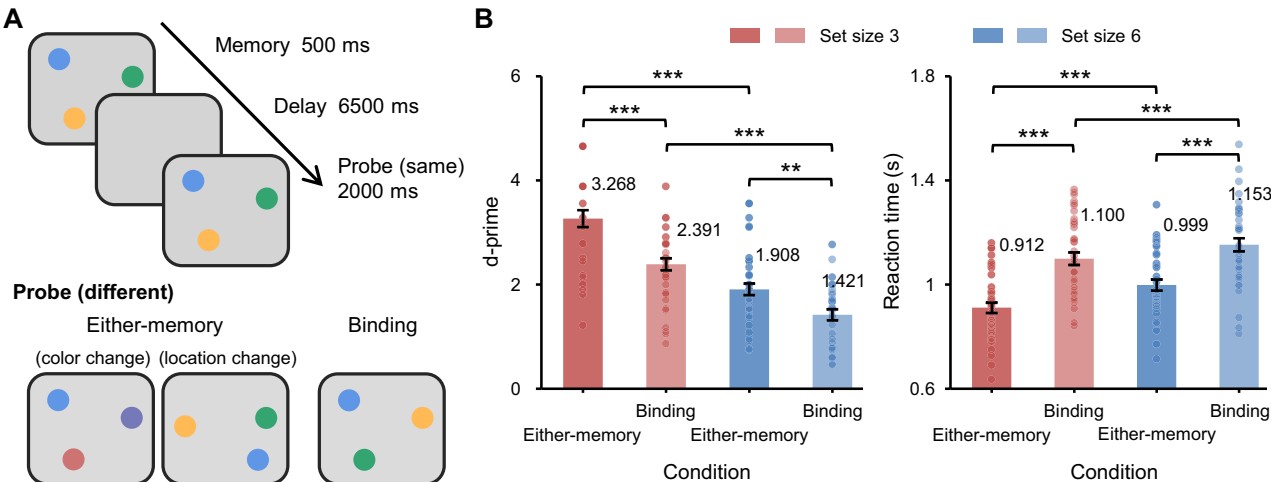

**Fig. 1 | Procedure and behavioral results. A** Participants were asked to perform a change detection task, which consisted of two conditions: (1) the either-memory condition, wherein either two colors or two locations were changed, with the task procedure remaining identical for both feature types up to the probe display; (2) the binding condition, wherein only the color was swapped between two disks, resulting in a modification in color-location bindings. Notably, in both conditions, participants had to maintain both color and location information throughout the delay period. The key difference was that in the binding condition, participants were required to encode and maintain the conjunction of color and location (i.e., bound representations), whereas in the either-memory condition, they could maintain the two features separately and only needed to retrieve the relevant one based on the probe. **B** Behavioral results of $n = 40$ participants for different conditions and set sizes. Each solid dot within the bar plots represents a participant, and the data variance is represented by ±1 s.e.m. **$p < 0.01$, ***$p < 0.001$.

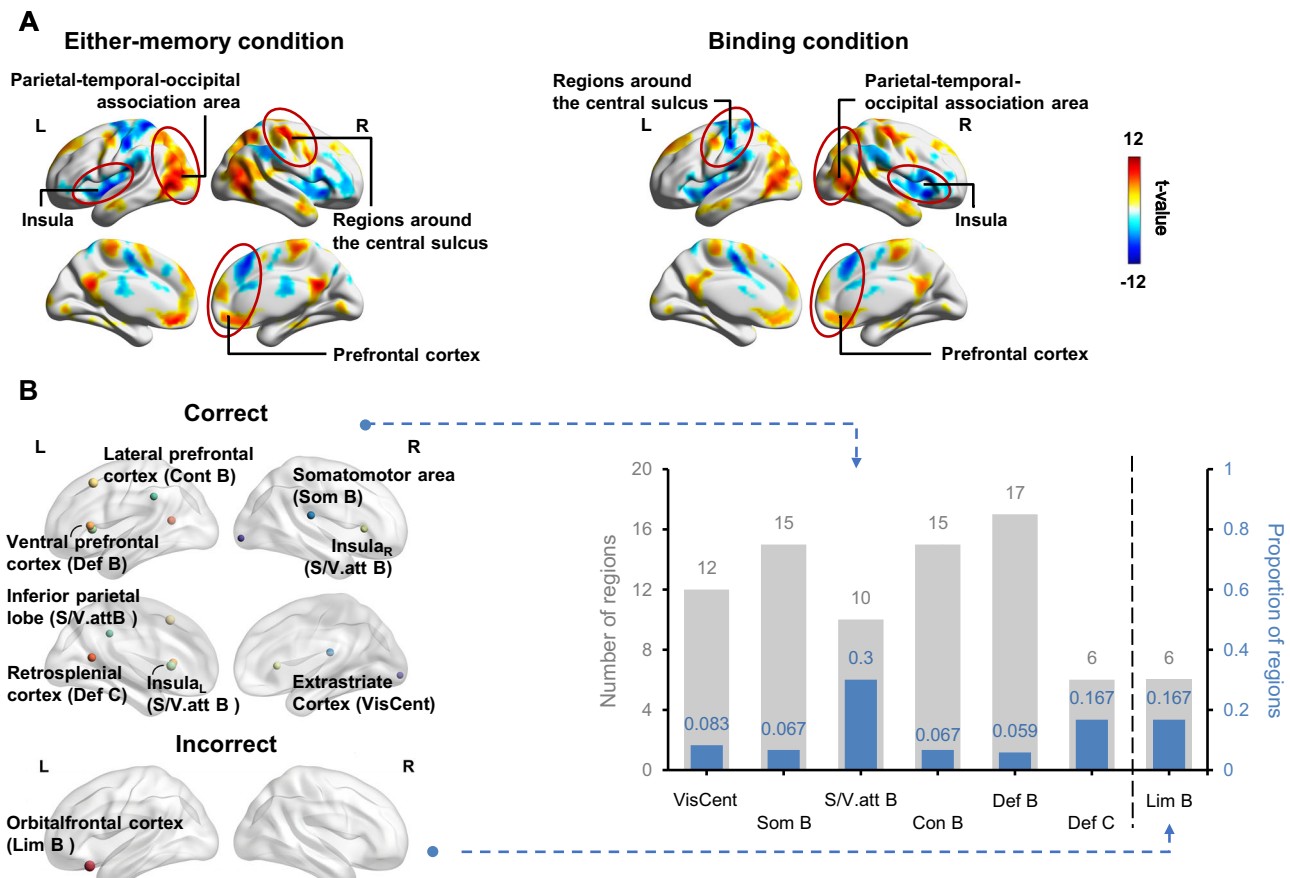

VisCent: Visual central network, Som B: Somatomotor B network; S/V.att B: Salience/Ventral Attention B network; Con B: Control B network; Def B: Default B network; Def C: Default C network; Lim B: Limbic B network.

**Fig. 2 | Univariate BOLD activity and local efficiency analyses. A** Neural activities (reflected by BOLD signals) for the either-memory and binding conditions, and their differences ($n = 40$ participants). Similar activity patterns were observed for these two conditions, prominently located in four brain areas (marked by red circles), involving the prefrontal cortex, insula, regions surrounding the central sulcus, and parietal-temporal-occipital association area. No significant difference was observed between conditions, $p < 0.001$, cluster-wise FWE correction. The color scale bar indicates t-values from the second-level GLM. **B** Left panel shows brain regions exhibiting significantly increased local efficiency in the binding condition compared to the either-memory condition ($n = 40$ participants), $p < 0.05$, FDR correction. Right panel indicates the proportion of these brain regions within their corresponding brain network. The gray columns represent the number of brain regions contained in each network, and the blue columns represent the corresponding proportion of brain regions within each network (e.g., 0.3 indicates that three out of 10 regions were detected with increased local efficiency in the Salience/ Ventral Attention B network for feature binding).

insula, regions surrounding the central sulcus (precentral and postcentral gyrus), and parietal-temporal-occipital association cortex (encompassing the inferior parietal lobe and middle temporal gyrus). Similarly, in the binding condition, the same activity patterns were identified, concentrated in these aforementioned areas ($p < 0.001$, cluster FWE correction), consistent with previous findings[14,27,28]. Yet, when comparing between conditions, no statistically significant differences in brain activity were observed (even when we reduced the statistical criteria to $p < 0.05$; same for analyzing different set sizes separately). This implies that, if there are any distinctions between the either-memory and binding conditions, they are likely to be minimal in terms of the raw BOLD signals. One possible reason for the observation of numerous brain regions involved in feature binding in previous studies may be that the number of to-be-memorized features was not equitably matched between conditions (feature-binding vs. single-feature).

Consequently, we assumed that the neural mechanisms governing feature binding are not exclusively attributed to a single brain region, but instead involve the collaborative contributions of a more extensive brain network and dynamic interactions between various regions, like other memory processes[25,26]. To test this hypothesis, subsequent analysis recruited graph-based network methods to construct functional brain networks to identify activities specifically associated with feature binding.

## Efficiency-based property

We recruited the Schaefer 200 parcel parcellation with 17 networks (MNI-3 mm)[29] to construct functional brain networks under the either-memory and binding conditions. Local efficiency ($E_{loc}$) was first calculated for each ROI (see "Methods" for details) to identify regions showing differences in the capacity for rapid and robust information transfer within the localized network[30] between conditions across the whole brain. The results revealed a significant increase in efficiency in the binding condition compared to the either-memory condition ($p < 0.05$, false discovery rate (FDR) correction). Specifically, this pattern was observed across eight brain regions for correct trials, involving extrastriate cortex (Visual central network), somatomotor area (Somatomotor B network), inferior parietal lobe (Salience/Ventral Attention B network), left and right insula (Salience/Ventral Attention B network), lateral prefrontal cortex (Control B network), ventral prefrontal cortex (Default B network) and retrosplenial cortex (Default C network). However, for incorrect trials, only the orbital frontal cortex (Limbic B network) was involved (see Fig. 2B). Statistical results are detailed in Table 1. These findings suggest that these regions are of greater importance in transmitting and processing information for feature binding. This provides preliminary evidence for collaborative contributions of various brain regions (centered around eight core regions) during successful feature binding.

**Table 1 | Increased local efficiency in the binding condition relative to the either-memory condition**

| Sign | MNI centroid coordinates | Name of Schaefer-200 template | Abbreviation | Networks | t-stat |
|---|---|---|---|---|---|
| Correct trials | (31, −94, −4) | R_VisCent_ExStr_3 | Extrastriate cortex | Visual Center | 3.323 |
| | (44, −27, 18) | R_SomMotB_S2_2 | Somatomotor area | Somatomotor B | 3.817 |
| | (−60, −39, 36) | L_SalVentAttnB_IPL_1 | Inferior parietal lobe | Salience/Ventral Attention B | 3.679 |
| | (−33, 20, 5) | L_SalVentAttnB_Ins_1 | Insula left | Salience/Ventral Attention B | 4.614 |
| | (36, 24, 5) | R_SalVentAttnB_Ins_2 | Insula right | Salience/Ventral Attention B | 3.632 |
| | (−40, 19, 49) | L_ContB_PFCl_1 | Lateral prefrontal cortex | Control B | 4.624 |
| | (−52, 22, 8) | L_DefaultB_PFCv_4 | Ventral prefrontal cortex | Default B | 4.015 |
| | (−11, −56, 13) | L_DefaultC_Rsp_1 | Retrosplenial cortex | Default C | 4.043 |
| Incorrect trials | (−24, 22, −20) | L_LimbicB_OFC_3 | Orbital frontal cortex | Limbic B | 5.020 |

### Extracted subnetwork of eight core brain regions

Although we identified eight brain regions (extrastriate cortex, somatomotor area, inferior parietal lobe, left and right Insula, lateral prefrontal cortex, ventral prefrontal cortex, retrosplenial cortex) that are crucial for feature binding, the specific roles and relative importance of these regions remain unclear. For instance, it is not yet clear which brain region dominates the functional connections during feature binding and in what manner these regions contribute to the process. To test this, we extracted a subnetwork from the $200 \times 200$ functional connectivity (FC) matrix for each participant, focusing on these brain regions. Subsequently, we applied network-based statistics (NBS) to explore neural evidence for regional collaboration within the subnetwork during feature binding (see "Methods" for details). The NBS detected a significant connected component involving seven brain regions (somatomotor area, inferior parietal lobe, left and right insula, lateral prefrontal cortex, ventral prefrontal cortex, retrosplenial cortex) and nine edges, favoring feature binding ($p < 0.05$, permutation test). Notably, the majority of significant edges originated from the somatomotor area (5), PFC (4), and left (3) and right (3) insula. Further examination confirmed that the connection weights of these edges showed a significant increase for the binding condition compared to the either-memory condition ($p < 0.05$, FDR correction; Fig. 3). These findings imply that conditional differences extended beyond the nodal level to the connectivity level, suggesting profound communication between these core brain regions during feature binding. Furthermore, the somatomotor area, PFC, and left and right insula acted as major contributors to the successful binding process.

### Intrinsic neural timescales of eight core brain regions during feature binding

Recent studies have used the intrinsic neural timescale to characterize functional hierarchy across the human and primate brain[31,32]. Here, we employed this method to identify where feature binding initially occurs and estimated autocorrelation function (ACF) values. Across eight core regions, the activity of somatomotor area exhibited the most rapid decay of ACF values with time (a larger slope before ACF hit zero), whereas other regions were correlated across slightly longer periods (Fig. 4A). Furthermore, we estimated the intrinsic timescale (sum of ACF values within the specific period; see "Methods" for details) for each brain region involved in the binding process. If a specific brain region has a significantly larger intrinsic timescale, its neural activity is likely to be more sustained and stable, enabling it to maintain robust representations over extended periods. Conversely, a smaller intrinsic timescale is associated with more rapid and transient neural activity, allowing the region to carry rich temporal information and respond rapidly with variations[32–34]. The results showed that the somatomotor area had a significantly shorter timescale than other regions (all $p$s < 0.05, FDR correction; Fig. 4B), which was not observed in the either-memory condition; while other regions exhibited a gradual increase in timescales. This hierarchy structure of intrinsic timescale suggests that the somatomotor area took part in feature binding in a more dynamic fashion and showed rapid responses to binding signals throughout the task, while

other regions processed binding information comparably steadily, albeit later in time.

### Correlated with behavioral performance

To determine the relationship between our connectivity findings and behavioral performance, a correlation analysis was conducted. Specifically, when correlating observed connection weights with mean RTs, we found significantly positive (Spearman) correlations for connections between somatomotor area and left insula, $r = 0.49$, $p = 0.002$, between somatomotor area and lateral prefrontal cortex, $r = 0.38$, $p = 0.017$, and between somatomotor area and ventral prefrontal cortex, $r = 0.49$, $p = 0.001$ (see Fig. 4C); but not for d-prime (see Fig. S1). That is, larger connection weights were associated with slower responses in the binding condition. Previous studies have shown that, when involving feature binding, extra binding processes produced larger RTs[15,35], which was also observed in the present study. This indicates that extra binding processes required stronger functional connections. In addition, these correlations also highlight the more important role of the somatomotor area in feature binding, as noted that, no such correlation was observed for other connections and those in the either-memory condition (see Fig. S2).

### Excluding the impact of task difficulty

Given that task difficulty differed between the either-memory and binding conditions, one might question whether the observed effects related to feature binding may have been confounded by task difficulty. To rule out this possibility, we distinguished different set sizes (reflecting different task difficulties) and examined their differences in local efficiency across eight core brain regions. The results revealed no significant differences for different set sizes (see Fig. S3).

To further rule out potential confounds related to task difficulty between the binding and either-memory conditions, we performed a control analysis with matched performance across conditions. Specifically, we selected participants with above-median performance in the binding condition and below-median performance in the either-memory condition, excluding three participants who fell into both groups. These two groups showed no significant differences in d-prime (binding: $M = 2.0$; either-memory: $M = 2.3$; $t(16) = 1.3$, $p = 0.202$) or RTs (binding: $M = 1.0$ s; either-memory: $M = 1.1$ s; $t(16) = 1.54$, $p = 0.133$), confirming comparable task difficulty. Reanalyzing FC within this matched sample revealed a significant increase in local efficiency for the binding condition across eight regions, including extrastriate cortex, somatomotor area, inferior parietal lobe, bilateral insula, lateral and ventral prefrontal cortex, and retrosplenial cortex ($p < 0.05$, FDR-corrected; Fig. S4). NBS further identified a significant component involving seven regions and nine edges ($p < 0.05$, permutation test; Fig. S5), with major hubs in the somatomotor area (5 edges), prefrontal cortex (4), and bilateral insula (3 each). These findings closely replicate the full-sample results, confirming that the observed connectivity differences are not driven by task difficulty but instead reflect mechanisms specific to feature binding.

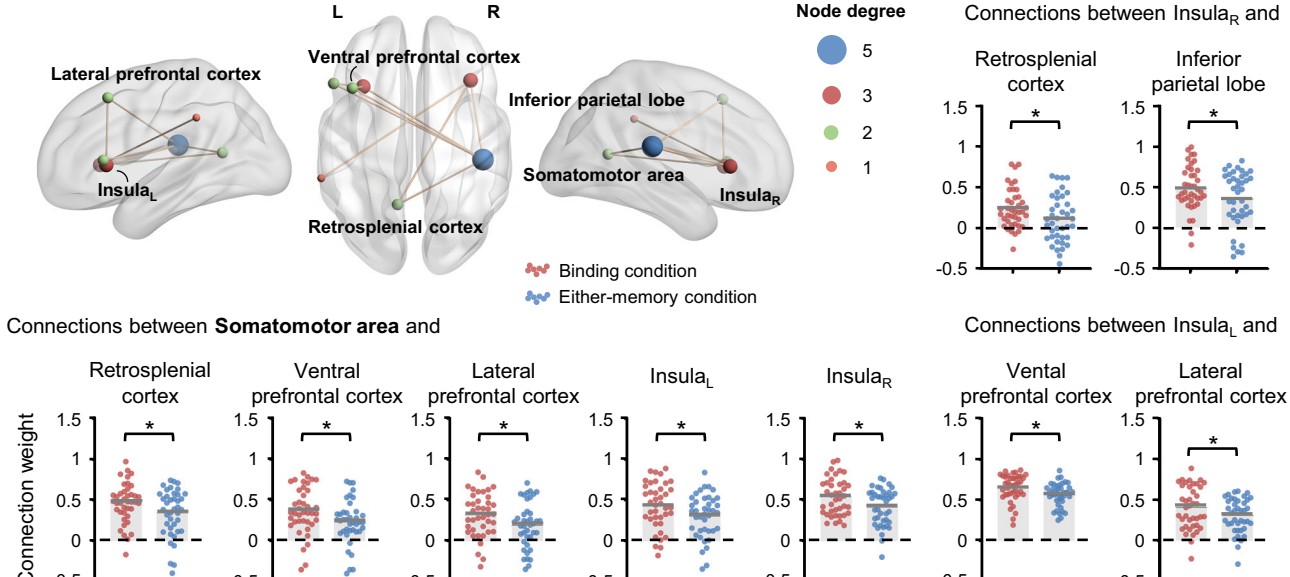

**Fig. 3 | The surface plot shows nine edges connecting seven brain regions that exhibit differences between the binding and either-memory conditions ($n = 40$ participants).** Node degree is defined as the number of edges directly connected to a specific region. The bar plots show the difference in connection weights for each edge between the binding and either-memory conditions, $*p < 0.05$, FDR correction. The horizontal bars indicate mean values, and each dot represents a participant.

In the end, it is important to note that the current results remained consistent when the Schaefer 400 parcel parcellation was used to construct FC network, and likewise when a fixed-density thresholding approach (at 10% sparsity) was applied to connectivity computing, validating the stability of the present observations (see Figs. S6–S8).

## Discussion

WM is acknowledged as a system capable of manipulating stored information for upcoming goals[36], albeit with a limited capacity[37]. Binding multiple features into a unitary entity in WM is crucial for enhancing its capacity to effectively support ongoing cognitive tasks. Because the neural mechanism underlying feature binding in WM remains unclear, the current study employed fMRI techniques and graph-based network analysis to systematically unravel the neural mechanism. The results indicate that compared to processing separate features, processing feature binding demanded additional attentional resources, as evidenced by relatively poorer performance (see also previous study)[9]. Simultaneously, we observed increased local efficiency in eight core brain regions involved in feature binding, including the extrastriate cortex, somatomotor area, inferior parietal lobe, left and right insula, lateral prefrontal cortex, ventral prefrontal cortex, and retrosplenial cortex. Subsequently, when constructing a subnetwork for these eight brain regions, we identified a connected component, encompassing nine significant edges across seven brain regions (excluding extrastriate cortex), contributing to feature binding. Overall, these results suggest that the neural mechanisms governing feature binding involve the collaborative contributions of a more extensive brain network. This network may function as a workspace[38,39] serving feature binding via dynamic interactions between these regions. The dedicated workspace exhibits not only better efficiency in information transfer across the eight brain regions, but also sufficient and valid internal connections for regional cooperation to facilitate successful feature binding in WM.

Among the eight core brain regions involved in feature binding, the somatomotor area, PFC, and left and right insula played critical roles in this process, as the majority of connections centered around them. Emerging evidence suggests a critical role for the somatomotor area in visual and attentional processing[40–43]. For instance, activity in this region can decode visual object categories[44] and modulate visuospatial attention[45,46]. Furthermore, studies in animals[47,48] and humans[49] have indicated the existence of connections between the somatomotor area and other higher-order cortices.

For instance, the somatomotor area is considered as a temporary storage site within the feedforward and feedback information transmission[50–52], and receives top-down signals from PFC and parietal cortex, directing upcoming actions[53,54]. These pathways enable the somatomotor area to integrate sensory, attentional, and motor information. This indicates that the somatomotor area might act as a core site integrating information from the visual, frontal, and parietal cortex to form early bindings in WM, reflecting online processing of sensory events and connectivity with other cortical and subcortical areas.

This was further underscored by the observed correlations with behavioral performance, where significantly positive correlations between connection weights and mean RTs were mainly observed for specific connections involving the somatomotor area (i.e., somatomotor area and left insula, somatomotor area and lateral prefrontal cortex, and somatomotor area and ventral prefrontal cortex), highlighting its important role and connections with other brain regions. Moreover, we observed a hierarchy structure of intrinsic timescales within the workspace, showing a gradual increase in timescales from the somatomotor area, ventral prefrontal cortex, retrosplenial cortex, inferior parietal lobe, extrastriate cortex, lateral prefrontal cortex, and left and right insula. The heterogeneity of the neural timescale indicates a possible sequence in information transfer between these eight regions. Human neuroimaging and macaque electrophysiology studies have shown that the heterogeneity of neural timescales serves as a basis of the functional hierarchy in the brain[55,56]. Considering this heterogeneity of intrinsic timescale, in the workspace for bindings, we propose that the somatomotor area, with a shorter intrinsic timescale, responds more rapidly to visual input. Subsequently, it relays information to regions with longer timescales (such as the insula and PFC) to establish a more robust way to process information. This makes the somatomotor area carry rich temporal information with more connections and positions it as the starting point during mnemonic feature binding. Similarly, insula, PFC, and other regions with relatively long timescales are more likely to receive information from the somatomotor area to establish stable representations of bindings and maintain them. Notably, the somatomotor area showed the strongest connectivity with other brain regions, rather than increased local activity (as reflected by BOLD signal differences). This suggests a dynamic, network-level role rather than static neural activity. Its specific functional contribution to feature binding, however, remains to be determined and requires further investigation.

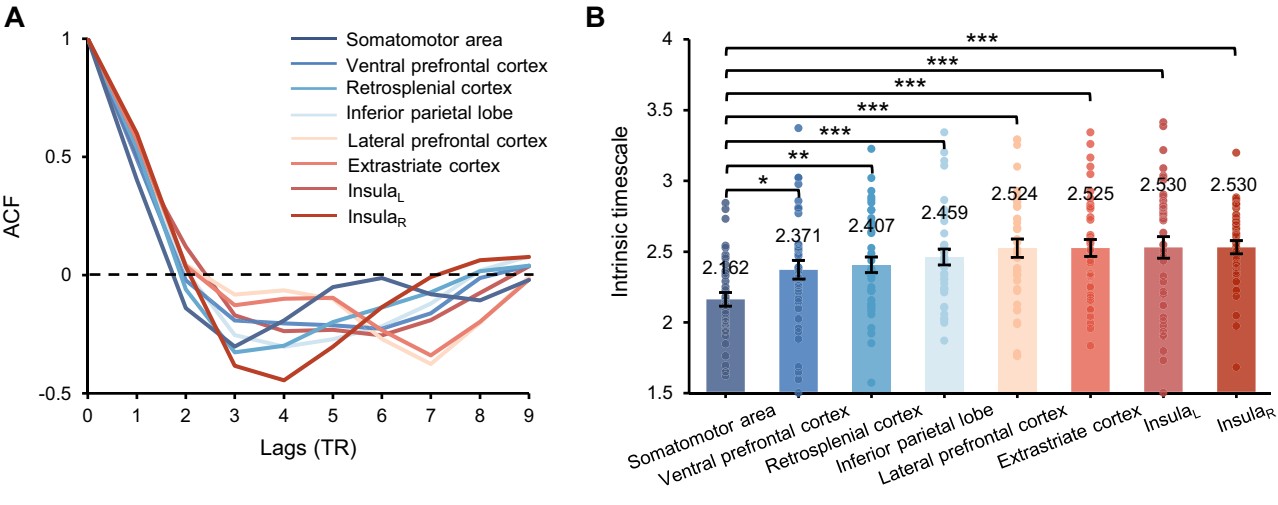

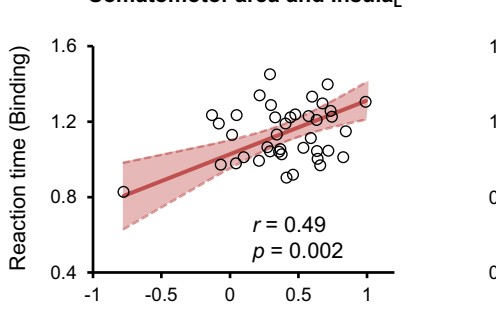

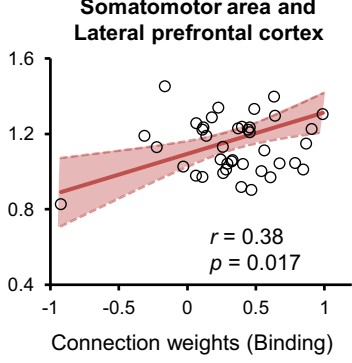

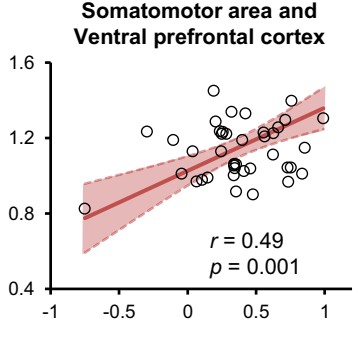

**Fig. 4 | Intrinsic neural timescales and connectivity-behavior correlations. A** The autocorrelation function (ACF) of eight core brain regions estimated during feature binding (TR 0 indicates the fixation onset). **B** The bar plot displays the intrinsic timescale (the sum of ACF values before it hits zero) of eight regions in the binding condition ($n = 40$ participants). The somatomotor area had a significantly shorter timescale than the other regions, * $p < 0.05$, ** $p < 0.01$, *** $p < 0.001$, FDR correction. Each solid dot within the bar plots represents a participant, and the data variance is represented by ±1 s.e.m. **C** Scatter plots indicate the correlation between connection weights and mean RTs for specific connections, between the somatomotor area and left insula, somatomotor area and lateral prefrontal cortex, and somatomotor area and ventral prefrontal cortex in the binding condition. Solid lines indicate linear fits to the data of $n = 40$ participants, with the shaded areas depicting the 95% confidence interval of the fitted lines.

Insular cortex, known as an integrative hub with extensive connectivity, receives sensory inputs from all modalities and establishes reciprocal connections with PFC and medial temporal lobe[57–59], linking information from diverse functional systems to ensure various cognitive functions, including episodic memory and salience detection[23,60,61], signifying its role in detecting feature binding. The PFC involved in feature binding is mainly attributed to its activation during information integration[62–64]. It has been shown that integrated representations from different feature dimensions are established through information exchange between different subdivisions of PFC[14]. Additionally, closely linked to cognitive control, PFC exhibits extensive connections to large-scale brain networks[65,66].

The involvement of the inferior parietal lobe and extrastriate cortex in feature binding aligns with the FIT[67] and relevant studies[68,69]. This theory emphasizes attention-dependent reentrant processes from the parietal cortex to the visual cortex as crucial in the binding process. Moreover, the specific activity of the parietal cortex has been observed in response to feature binding within WM[15,18,70] and spatial attention[27,35]. The retrosplenial cortex, situated at the intersection of areas encoding visual information, motor feedback, and higher-order decision making[71–73], is in an ideal position to integrate these inputs, especially related to spatial attention[74–76]. The close correlation between retrosplenial cortex activity and spatial attention further underscores its relevance to feature binding.

Feature binding in WM has been extensively studied using change detection tasks, where participants compare an initial stimulus array with one presented after a short delay[8,18,19]. Employing these tasks, studies have implicated the hippocampus[11–13], prefrontal cortex[14], parietal cortex[13,15], and primary visual areas[16,17] in feature binding. Extending this work, the present study highlights additional involvement of the somatomotor area and insula in binding processes. A key limitation of earlier studies is the frequent comparison between binding and single-feature memory conditions. This can confound binding-specific neural effects with differences in feature load. To address this, we employed a more stringent comparison: a binding condition versus an either-memory condition. In both, participants memorized both color and location information. However, only in the binding condition was it necessary to encode and retain their conjunction. In the either-memory condition, features were stored separately, and only one was retrieved based on the probe. This design ensured that observed neural differences reflected binding demands rather than memory load or task difficulty (see also our performance-matched control analysis). Moreover, rather than focusing solely on univariate activation, we used graph-based network modeling to examine large-scale neural coordination. This revealed that feature binding relies on enhanced local efficiency and interregional connectivity among the somatomotor area, insula, and prefrontal cortex. These findings suggest that binding is supported by distributed processing within a central neural workspace, extending current models of WM. Our results thus offer empirical support for the dynamic, network-level mechanisms underlying feature binding.

Importantly, the present study utilizes a task variant that preserves the diagnostic role of spatial location while imposing multi-item interference at probe[8,9]. This design permits a stringent test of conjunction maintenance under matched memory load. Nonetheless, alternative task variants, such as single-item probes, may differ in the degree of probe interference or retrieval demands[19]. Our findings should therefore be interpreted as reflecting network-level coordination within this specific task context, rather than a general signature of binding across paradigms. Future work directly comparing binding variants within the same imaging framework will be essential for determining which neural effects generalize across tasks and which remain task-dependent.

It is noteworthy that our conditional difference analysis in BOLD signals did not reveal any significant brain regions supporting feature binding, which is a discrepancy with previous studies[13,14,18]. This variance may arise from the specific contrast made between the binding and either-memory conditions in the present study. Unlike other studies that might have additional differences in the number of to-be-memorized features, our focus was solely on the process of binding itself. In addition, the necessity of correcting for multiple comparisons in both voxel intensity and cluster size in voxel-level analysis enhances the stringency and reliability of the results. However, this rigorous correction procedure might diminish the sensitivity of voxel-level analysis, making it challenging to detect subtle effects or small ROIs[77]. The crucial point, though, is that despite the lack of significant findings in BOLD signals at the voxel level, the connection weights for significant edges in the subnetwork were associated with participants' behavioral performance. This suggests that feature binding involves a more complex brain network with extensive interactions between various regions, and such subtleties might not be captured by traditional GLM analysis alone. The graph-based network analysis employed provides a more nuanced understanding of the collaborative contributions of brain regions involved in feature binding.

The feature binding task we used required participants to integrate visual features (color) with spatial locations, engaging both perceptual and spatial attention mechanisms. This aligns with classical feature binding theories[2] that emphasize spatial attention's role in combining separate features into unified objects. Our results showed that the binding condition elicited significantly greater connectivity in regions including the somatomotor area, insula, and prefrontal cortex compared to either-memory conditions (which involved only visual or spatial memory). These regions were not similarly activated in spatial-only or visual-only tasks, indicating that they reflect integrative control processes rather than the additive effect of individual feature maintenance. This suggests that binding demands recruit additional neural mechanisms beyond those needed for visual or spatial memory alone. Thus, although the task depends on visuospatial representations, the binding condition uniquely engages higher-order neural networks supporting feature integration. These findings extend Treisman's framework by highlighting the involvement of distributed neural systems in feature binding beyond early sensory or spatial-specific regions.

Gaining insight into feature binding and the associated neural mechanisms is important for different models of feature binding. It is reasonable to conclude that there are specific neural mechanisms dedicated to feature binding, with a central workspace involving somatomotor area, insula, and PFC, underpinning the effective processing of bindings. Notably, the somatomotor area is highlighted for its pivotal functions in connecting with other regions and serving as the initial point in the feature binding in WM. This idea of a dedicated workspace implies a collaborative network where these brain regions dynamically interact to facilitate successful feature binding, providing valuable insights into the neural underpinnings of this cognitive process.

## Methods
### Participants
Forty right-handed volunteers (18 males, aged $19.45 \pm 1.21$ years) were recruited from South China Normal University. Sample size was predetermined to be approximately twice that of comparable neuroscience studies investigating feature binding (with sample sizes ranging from 5 to 22)[11,16,18], and 40 participants would be enough to replicate the previous behavioral findings[9] with a power larger than 0.99. All participants had normal or corrected-to-normal vision, and none had a history of neurological or cognitive disorders. Prior to testing, participants underwent screening for MRI contraindications and provided written informed consent. All participants were compensated for their time, and the experimental procedures were approved by the Institutional Review Board of South China Normal University (2020-3-013), and all ethical regulations relevant to research involving human participants were strictly followed.

### Stimuli
Three or six colored disks (with a radius of 1.25°) were presented against a gray background. Their colors were randomly selected from nine colors with equal brightness and evenly distributed along a circle in the CIE Lab color space (centered at $L = 70$, $a = 5$, $b = 0$; with a radius of 40). Their locations were randomly selected from a three-by-three invisible grid around the screen center (excluding the center itself), with each grid subtended by $4° \times 4°$. To prevent attentional adaptation, the location of each disk was randomly jittered by $\pm 0.64°$ in the vertical direction on each trial.

### Procedure and design
Figure 1A illustrates the sequence of events for a single trial. Each trial began with a white fixation dot presented for 500 ms, followed by a memory display including three or six different colored disks (randomly mixed within each run) for 500 ms. After a delay of 6500 ms, the test display appeared for 2000 ms. Participants were asked to indicate whether the test display had changed compared to the memory display by pressing the "4" (change, 50% trials) or "3" (no change, 50% trials) button. Feedback was provided for 1000 ms, and the next trial started after a randomly chosen inter-trial interval (ITI) lasting 4500, 6000, or 7500 ms. Consistent with previous studies that reliably reported feature-binding–related activations[11,14,18], we did not require participants to maintain strict fixation throughout the trial, as eye movements do not disrupt the binding process[3,78].

The experiment comprised two conditions: (1) The either-memory condition, wherein either two colors or two locations were changed to the new ones with an equal probability in the "change" trials. Participants had to detect and report these changes, with the task procedure remaining identical for both feature types up to the probe display. (2) The binding condition, wherein only the color was swapped between two disks in the "change" trials, thereby altering the color-location bindings. Importantly, in both conditions, participants had to retain both color and location information throughout the delay period. The key difference was that in the binding condition, participants were required to encode and maintain the conjunction of color and location (i.e., bound representations), whereas in the either-memory condition, they could maintain the two features separately and only needed to retrieve the relevant one based on the probe.

Different conditions were tested in different fMRI scans. For each condition, participants completed a practice run consisting of 16 trials before fMRI scanning, and three runs of 24 trials during fMRI scanning. Different set sizes were mixed within each run, and the conditions were counterbalanced between participants.

### fMRI data acquisition
Functional images were acquired using a 3 T Siemens Trio scanner with a 20-channel phase-array coil at the Brain Imaging Center in South China Normal University. Stimuli were projected onto a translucent screen with a refresh rate of 60 Hz and a spatial resolution of $1024 \times 768$. Participants observed the stimuli through a mirror positioned above their eyes at a distance of 92 cm.

For the acquisition of echo-planar imaging (EPI) images, a gradient EPI sequence with a TE (echo time) of 30 ms and a TR (repetition time) of 1500 ms was employed. The EPI sequence comprised 46 axial slices with a

3 mm slice thickness, a $64 \times 64$ acquisition matrix, and a field of view (FOV) of $192 \times 192$ mm. High-resolution anatomical images were also acquired for each participant using a T1-weighted MPRAGE sequence with a TE of 2.27 ms, a TR of 1900 ms, a flip angle of 7°, and an FOV of $256 \times 256$ mm. The MPRAGE sequence featured an isometric voxel resolution of 1 mm and covered the whole brain with 208 sagittal slices.

Frame-wise displacement (FD) was computed for participant exclusion[79]. Participants exhibiting high levels of motion, defined by a mean FD greater than 0.5 mm, were excluded. No participants were excluded due to excessive movement during the MRI scans.

## Data preprocessing and general linear model
SPM12, version 7771 (https://www.fil.ion.ucl.ac.uk/spm/software/spm12/), was used for the analysis of the imaging data. All EPI images were slice-time-corrected, motion-corrected, registered to the Montreal Neurological Institute (MNI) template, spatially smoothed (with an isotropic 6 mm full-width at half-maximum Gaussian filter), and high-pass filtered (128 s).

For each condition, a first-level GLM was applied individually on the preprocessed functional data. Event-related regressors were obtained by convolving the onset of each trial with the canonical hemodynamic response function, capturing the expected BOLD response. Each entire trial—from onset to ITI—was modeled as a single regressor to maximize sensitivity to sustained activity[80]. And six head-movement-related regressors were also included in the model to account for motion artifacts. Subsequently, the second-level GLM was performed to complete a group analysis, calculating brain activations (reflected by statistical maps based on t-values) for each condition, and the contrast between conditions. The threshold of statistical maps was set to $p < 0.001$, and corrected using cluster FWE correction.

## Graph-based network analysis overview
Overall, since we hypothesized that the neural mechanism underlying feature binding is attributed to collaborative contributions of various regions, graph theory, which specializes in detecting intricate connectivity patterns and tracking their dynamic changes, was employed independently to construct and compare functional brain networks between the either-memory and binding conditions. Initially, we constructed static functional brain networks and examined differences in the efficiency-based network properties (i.e., local efficiency) between conditions. Brain regions showing significant differences were identified, and a subnetwork was formed based on these regions. Next, NBS[20] were applied to pinpoint the different relationships across distinct brain regions between conditions. Additionally, the intrinsic neural timescale was calculated to explore the potential hierarchical structure of regions. Moreover, we established correlations between the observed network evidence and participants' behavioral performance. The Brain Connectivity Toolbox (https://sites.google.com/site/bctnet/)[81] and BrainNet Viewer (https://www.nitrc.org/projects/bnv/)[82] were adopted for these analyses.

## Static functional network
The Schaefer 200 parcels with 17 networks (MNI-3 mm)[29] were adopted to define the nodes (ROIs). To estimate FC between these nodes, time series were first computed for each node (ROI) by averaging the BOLD signals across all voxels within the respective node (ROI). Subsequently, (Pearson) correlations between each pair of nodes in the time series were calculated, resulting in a $200 \times 200$ FC matrix for each participant, representing weighted edges between nodes (ROIs).

To ensure the network was fully connected, we employed a hybrid two-step procedure combining the strengths of the minimum spanning tree (MST) and fixed-density thresholding, an approach increasingly adopted in the literature[83,84] resulting in a binarized functional matrix (unweighted) with 200 nodes and 199 edges. MST ensures network connectivity and topological consistency, while subsequent sparsity-based edge addition allows for meaningful computation of segregation and integration metrics. This combination mitigates the fragmentation risk inherent in thresholding alone and the oversimplification risk of standalone MST.

Step 1: MST as a backbone for connectedness and topological validity.

We used MST strategically as an initial backbone, guaranteeing a connected and acyclic structure ($N-1$ edges) for each participant's network. This step ensures that graph metrics—such as local efficiency—are well-defined and comparable across participants by eliminating isolated components.

Step 2: Sparsity-based densification to restore local structure.

We incrementally added the strongest edges until reaching a fixed sparsity threshold (10%)[84], reintroducing local connectivity patterns. This densification allows valid calculation of segregation metrics while maintaining global comparability. Importantly, the final networks remain fully connected and matched in edge density across participants—crucial for group-level analyses.

Notably, sparsity-based thresholding is widely used in network neuroscience to balance inclusion of biologically meaningful connections and exclusion of weak or spurious correlations[20,85]. The 5–20% range is well validated, and our choice of 10% aligns with standard practice, facilitating comparability with prior studies[84].

Finally, by multiplying the binarized network matrix with the original weighted connectivity network, a weighted connectivity matrix with 10% sparsity, comprising 200 nodes and 1990 edges, was obtained for each participant.

## Efficiency-based network properties
For each type of trial (correct vs. incorrect), efficiency-based properties (i.e., local efficiency) were computed separately for the binding and either-memory condition across different ROIs. Specifically, local efficiency ($E_{loc}$) signifies the ability for parallel information transfer within a neighborhood, which is the immediate neighbors of each node within the network[30]. Given a graph $G = (V, E)$, let $v$ be a node (i.e., a fundamental unit that can indicate a region in the brain network) and $e$ be an edge (i.e., pairwise connectivity between nodes) in the graph. $N(v)$ denotes the set of neighbors of $v$ (i.e., directly connected to $v$). Then, $G[N(v)]$ represents a subgraph comprised of the neighbors of $v$ and all the edges (i.e., connections) between these neighbors in $G$. The $E_{loc}$ of node $v$ was defined as

$$E_{loc}(v) = \frac{1}{|N(v)|\left(|N(v)| - 1\right)} \sum_{i,j \in N(v), i \neq j} \frac{1}{d_{ij}},$$

where $|N(v)|$ is the number of neighbors of $v$, and $d_{ij}$ is the minimum number of edges that must be traversed to travel from node $i$ and node $j$ in the subgraph $G[N(v)]$. The higher the efficiency of a brain region, the better it can transmit and process information, indicating its greater importance in ongoing activities. T-tests and FDR correction were used to detect differences in local efficiency between conditions. Subsequently, brain regions exhibiting significant differences in nodal local efficiency between conditions were employed to extract the subnetwork from the $200 \times 200$ FC matrix for each participant.

## Network-based statistic (NBS)
In the extracted subnetwork, the edges (representing pairwise connectivity between nodes) were compared between the binding and either-memory conditions using t-tests, generating a matrix of t statistics. By setting the threshold at $p < 0.05$, a group of significant edges was identified, where a stronger connection in these significant edges indicates a more robust synchronization between brain regions. NBS was then applied to detect the number of significant edges that form connected components. Connected components are sets of nodes where all pairs are connected by edges. The size of these components, denoting the number of edges, was determined. A permutation test with 10,000 random iterations was employed to assess the significance of the identified connected components based on their size, while maintaining the original threshold for $p < 0.05$. For NBS, it is notable that the null hypothesis was consistently rejected at the component level, rather than at the individual edge level.

## Intrinsic neural timescale

This approach aimed to evaluate the duration of neural information storage in different brain regions[31,32]. Averaged BOLD signals across trials were used to estimate an ACF[86] for each brain region under each condition. Afterwards, the sum of ACFs within a specific period was used to estimate the intrinsic neural timescale[33,87]. This specific period was defined as the area under the ACF curve up to the time lag value just before the ACF became non-positive for the first time with increasing time lag. To correct the potential variants in temporal resolution, the sum of ACFs under the identified area was then multiplied by the TR. Noted that, a larger intrinsic timescale signifies neural responses with variations over longer intervals, associated with sustained and stable neural activity; while a smaller intrinsic timescale indicates neural responses over shorter intervals, often linked to a more rapid and transient neural activity[34].

## Statistics and reproducibility

Statistical analyses were conducted in MATLAB v.2018 (MathWorks Inc.) together with SPM12 and established toolboxes for network analysis. Unless otherwise specified, all statistical tests were two-tailed, and significance was assessed at $p < 0.05$. For voxel-wise fMRI analyses, statistical maps were thresholded at $p < 0.001$ and corrected for multiple comparisons using cluster-level FWE correction. For nodal network measures, condition differences were evaluated using paired t-tests with FDR correction. NBS were assessed using nonparametric permutation testing with 10,000 iterations, and significance was determined at the component level.

The primary unit of analysis was the individual participant. Forty healthy adults were included in the final analyses, and no participants were excluded following motion quality control. Each participant completed both experimental conditions, enabling within-subject comparisons. No trial-level repetitions were used to estimate individual-level effects and were not treated as independent observations.

To assess the robustness and reproducibility of the findings, key analyses were repeated using alternative network construction strategies, including a higher-resolution parcellation (Schaefer 400 parcels) and a fixed-density thresholding approach (10% sparsity). The main results remained consistent across these analytic choices, supporting the stability of the reported effects.

## Data availability

The anonymized fMRI data supporting the findings of this study will be made available at https://osf.io/gcu68/ (ref. 88).

## Code availability

The code we used can be accessed through https://osf.io/gcu68/ (ref. 88).

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

## Acknowledgements

This research was supported by the National Science and Technology Innovation 2030 Major Program (grant no. 2021ZD0200534 to B.W.), the National Natural Science Foundation of China (grant no. 62303143 to H.W.), and the National Social Science Major Program (grant no. 20&ZD296 to X.W. and B.W.). The funders had no role in study design, data collection and analysis, decision to publish, or preparation of the manuscript.

## Author contributions

Y.C., F.C., and B.W. designed the experiment. Y.C. collected the data, Y.C., F.C., and H.W. analyzed the data. Y.C., H.W., J.H., X.W., and B.W. wrote the paper. All authors approved the final version of the manuscript for submission.

## Competing interests

The authors declare no competing interests.
