## [Transparent Peer Review file · Communications Biology]

Neural mechanisms of feature binding in working memory

Corresponding Author: Dr Yang Cao

Version 0:

Reviewer comments:

Reviewer #1

(Remarks to the Author)

In this study, the authors investigated the neural processes governing feature binding in working memory using a change detection task. Subjects had to remember either the colour, location, or both of simple disks on the display. The condition in which subjects had to remember was the feature binding condition. Subjects performed this task while having their neural activity scanned in an fMRI scanner. The authors used graph-based network analysis to examine the functional activations of cortical network governing feature binding in working memory. Behavioural performance was found to be better – i.e., higher d-prime and faster reaction times - for smaller set sizes and the conditions testing only either location or colour. Their main fMRI findings show that within cortex-wide network governing both “either-memory” and “binding memory” conditions the latter was associated with enhanced local efficiency. Thus, the fMRI data also reveal specific brain regions that play a key role in feature binding in working memory.

The manuscript is very well-written. The study is novel and very interesting. The results could have a significant impact on our understanding of the cortical mechanisms underlying working memory and feature binding. I have some general comments and then some minor ones. I hope the authors find my feedback helpful in improving their manuscript. Given my comments I recommend major revisions.

- It appears as though the authors combined the colour only and location only change trials together as the “either-memory condition” to compare with the “binding condition”. I wonder why the authors did not conduct their statistical analyses on all three conditions: 1 binding condition and two separate either-memory conditions (one for colour and one for location)? In fact, combining both change conditions of only one feature doubles the number of trials relative to the binding condition. Would be nice to either show that the differences between colour only and location only conditions were statistically different or not. If they are not different, then that could be justification for combining them although I still think that is better to keep them separated. Moreover, I wonder how the neural data would differ between the location-only and colour-only conditions. There could be important differences in the neural processing between these two conditions. I feel like there is more that can be revealed by separating out the either-memory conditions.
- Can the authors please be more precise in their language when stating the meaning of the local efficiency analyses on page 8 where they say “...showing different information transfer abilities...” Precisely what do they mean by “abilities”? I don’t mean to nitpick here, but this is too vague for me and I don’t believe that readers with less experience in this form of analyses will really understand what is meant here. Please be more precise.
- Can the authors please clarify in the procedure section how the “either-memory” conditions changed either the colour or the location? In the Binding condition it appears as though two coloured disks swap locations as the other stays unchanged in location and colour. I suppose that is true for other coloured disks in that condition? But according to Figure 1A, the either-memory condition for colour appears to show that multiple disks can change colour? Likewise, in the either-memory condition for location appears to show that all the disks change location?
- Following my last comment and if I am correct, what that seems to me is that these either-memory conditions are exceedingly easier than the binding condition. The authors do address the idea that there might be potential differences in task difficulty between the either-memory condition and the binding condition. However, when addressing this point they only consider comparing set-size conditions. I believe that is not the correct comparison. Really, task difficulty is in the nature of the changes themselves and that the either-memory condition ... really conditions.... are far easier than the binding condition. I am not entirely convinced that task difficulty did not confound the results.

Minor comments:

- When citing papers the authors are inconsistent on whether or not they include first initials of the paper’s authors. Usually,

the initials are not included in the text.

• Page 8: "These findings suggest that these regions are of greater important in transmitting and processing information for feature binding." Appears to be a typo and should read maybe something like "...regions are of greater importance in transmitting..."?

Reviewer #2

(Remarks to the Author)

In their study "Neural mechanisms of feature binding in working memory" Drs Cao and colleagues use fMRI and network analysis to identify a network associated with binding mechanisms in working memory (WM). While this is an important topic and the work is based on a large set of data, I have three major concerns.

1. The WM conditions, called either-memory and binding conditions (both based on the well-established change detection task), show a confound that is potentially crucial. The either-memory condition tests whether participants notice changes in color or spatial location in the memory probe relative to the memory array shown earlier. For both types of changes global strategies can be used. Specifically, a color change can be detected based on ensemble perception and global color statistics. Location changes can be detected based on changes in the gestalt of the memory array vs. probe display. In contrast, in the binding condition memory items swap color such that binding color and location is necessary to find a change. Unfortunately however, that means that global strategies of color statistics or gestalt processing will no longer work. Furthermore, the difference in task difficulty could lead to participants using more effort and/or being more alert. It is difficult to estimate whether any results reported in the study are due to differences in used strategies, effort, or the presence/absence of binding.

2. Although I'm not an expert, it seems that the data analysis is somewhat unusual. From what I understand there is a certain risk with using Prim's minimal spanning tree algorithm (MST). It can simplify connectivity too much, deleting connections that are biologically relevant. It seems that this method tends to be used in a complementary fashion together with other approaches that were not used in the current manuscript. In addition, adding edges until a sparsity criterion is reached is, although not incorrect, somewhat unusual as it blends two rather different ideas of network analysis. Furthermore, local efficiency for the tree structure obtained via MST would be zero. Only with the added edges this calculation becomes meaningful. But then it is heavily dependent on which sparsity criterion is used where a sparsity criterion of 10% might be somewhat arbitrary. All this brings me to the question, why not using a thresholding approach to begin with or two approaches?

3. The finding that the somatomotor area plays a central role in binding is very surprising. The area is not known to be involved in visual functions and although it might be associated with some sort of goal-directed behavior I'm having great difficulties imagining how the current change detection task (that merely requires yes/no responses) would trigger any spatially specific action planning. Although finding a result unusual in and of itself is not necessarily a strong argument. But given that the current network analysis is unusual as well and that the somatomotor area doesn't show an involvement in the standard event-related fMRI analysis (Fig. 2A), I remain quite skeptical.

Additional comments:

Abstract and elsewhere. The abbreviation 'SMA' is almost always used for the supplementary motor area which is entirely different from the area referred to here. I would recommend using a different abbreviation to avoid confusion.

Introduction, 1st para, last sentence. ..." binding often results in a general deterioration in memory accuracy when it comes to integrating features"

Please provide a reference. Also, it seems there are fewer studies that examine WM accuracy as opposed to WM precision.

Figure 1, Behavioral performance & Methods: given the above-mentioned concerns about confounds, it seems not surprising that the binding condition yields significantly lower d-prime values.

Figure 2. Contrast between conditions. The brains seem to show no numerical difference (or was something lost in my copy of that figure?). Perhaps it's not necessary to show those brains?

Reviewer #3

(Remarks to the Author)

In this study by Cao et al., cortical mechanisms underlying working memory mechanisms for feature binding were investigated. Using univariate approaches, the authors found specific regions involved in both the feature binding and single-feature tasks. In addition, graph theory approaches were used to supplement how feature binding might related to intricate functional networks, which they found to be the case, with SMA as a key region. This is a timely study investigating the mechanisms for feature binding, especially using approaches that reflect the complex functional networks that have been little addressed to date.

Comments:

1. Could the authors please mention, even briefly, the task in the abstract?
2. When relating the current study to previous studies on feature binding, where specific cortical regions have been found, could the authors mention them explicitly? Then, could the authors compare those studies from a methodological standpoint (esp. if they are also visuospatial as in this task) and also, how their cortical findings compare to the current study?
3. Could the authors comment on cortical activity for trials that were correct - do the same regions appear using univariate and graph theory approaches?
4. While the authors mention that an initial trial fixation occurred, could the authors clarify whether fixation was maintained throughout the trial - I wonder what the link between SMA and potential several eye movements being made during the key phases?
-In related studies, were eye movements allowed and, if not, how was this reflected in cortical data?
5. In terms of the analysis, did the authors take the entire trial as a regressor, or were individual phases of the trial parsed out? Additional detail regarding this would be helpful.
6. Could the authors comment on the fact that this feature binding task was visuospatial in nature and how the results in the feature binding condition relate to the activation in either the spatial only task or the visual feature only task?
7. How do the regression plots look for connection weights and d-prime, especially for the plots shown in 4C?
8. Minor: When referring to insular cortex without 'cortex', the authors may refer to it as 'insula'.

Version 1:

Reviewer comments:

Reviewer #1

(Remarks to the Author)

I remain unconvinced by the authors' responses. As I stated in my first set of comments, and like Reviewer 2, I am skeptical about what the tasks is actually testing and what the results are actually reflecting. I am unconvinced that there is any difference in memory engagement between the two conditions. Subjects are required to retain both location and colour, regardless of condition. What is really different between the two conditions is how subjects' memory is tested - i.e., one condition is easy and the other is difficult. Reviewer 2 was more explicit than I was about what strategies subjects might employ when performing the task. My view is aligned with Reviewer 2. I am unclear with how the authors attempted to address our comments. It simply didn't make sense to me. I know fMRI studies require a lot of work and I am sorry I cannot be more positive.

Reviewer #2

(Remarks to the Author)

I would like to thank the reviewers for considering my concerns.

Concern 1:

As for my first concern of a confound I pointed out that the either-memory and binding conditions could be affected by confounds. The authors addressed one part of my concern regarding task difficulty. It is a good sign that contrasting set size as an experimental manipulation of difficulty did not yield results comparable to the main results of the study. An additional analysis seems less convincing – although it might as well stay in the manuscript. It used a median split of participants according to their either-memory or binding performance. I didn't quite understand how this split was done as the split was two-fold (according to better/worse performance in either of the two conditions) which could in principle result in four subgroups but only two groups were analyzed. Alternatively, the two splits could have been done separately but then partially the same participants would belong to the groups of "good binding" and "poor either condition" performers which would be inappropriate. If the authors could clarify which of the two approaches (or a different one) was used. Regardless of how the split was done, it does not create new independent variables but extracts person-based information about a mix of interindividual abilities, strategies and randomness. It doesn't hurt to include the median-split analysis but the manuscript should probably comment on the caveats that come with such limits. Once again, I felt that the set-size analysis was already helpful.

The other part of my concern pertained to the possibility that the either-memory and binding conditions might have involved very different brain functions. I appreciate the authors line of argument that different strategies should result in differences in task difficulty and that was already ruled out as an explanation of the current results. Unfortunately however, I am not convinced by that response. Neither do different task strategies have to necessarily differ in difficulty, nor is there only one kind of task difficulty (see Nili Lavie's perceptual vs. cognitive load to name just one example).

To reiterate my concern: in the either-memory condition participants probably pursued global ensemble strategies of extracting information about the gestalt of the dots in the memory array and about the first/second/etc. order statistics of its color distribution – and subsequently retaining such information to detect changes. Brady and Alvarez (e.g., 2011; 2015) have shown that participants use such hierarchical strategies. Furthermore, there is evidence to suggest that being *unable* to use such strategies can result in qualitatively different performance. My concern is that qualitatively different functions might be involved in the binding condition because it prevents global strategies. That is, neither the spatial gestalt nor the color stats of the memory array ever change in the binding condition. The problem would be that the contrast in the current study could potentially compare vastly different brain functions.

Perhaps there is a way of arguing that vastly different strategies should have resulted in different networks whereas the updated Figure 2 shows substantial overlap for the two conditions. This makes me wonder whether it is possible to include a

control analysis that first extracts voxels in a manner of a conjunction analysis, then conducts the graph-based analysis (i.e., for both conditions together) before back-projecting the results onto the two conditions. But for now I am not convinced that my concern about a confound is removed.

Concern 2:

Thanks for the additional clarifications and the extra figure. It is good to hear that a different approach yields similar results thereby demonstrating that the findings are robust. I might have missed it but I didn't find the additional analysis in the manuscript. If it is not included yet then perhaps it could be added to the supplementary material.

Concern 3:

The authors have clearly put significant thought into addressing my third concern. The authors point out that both their experimental conditions show strong involvement of the somatomotor area. However, there are alternative explanations for this activation to do with salience or attention, rather than the formation of visual feature conjunctions. The main difficulty, however, is that there is no differential activation for the two experimental conditions. The cited papers by Frost & Metin (1985), Sieben et al. (2013), Sun et al. (2016), and Taylor-Clarke et al. (2002) show an involvement of the somatomotor area, but their tasks always involve both visual and somatosensory processes. Smith & Goodale (2015) used a purely visual task and found that object classes could be decoded from S1 and S2. This likely reflects top-down projections from high level ventral and/or dorsal areas, or from prefrontal cortex, consistent with the reciprocal connections between the somatomotor area and higher-order regions (Morecraft et al., 2012; Rolls et al., 2023; Shipp et al., 1998). However, such top-down projections don't amount to evidence for binding. In sum, the present findings might involve attentional or saliency functions, they might also involve classification signals reverberating from visual areas but they do not demonstrate conjunctive coding. To demonstrate conjunctive representations it would be important to demonstrate that conjunctive information can be decoded from the somatomotor area (e.g., Pollmann et al, NeuroImage, 2014).

Reviewer #3

(Remarks to the Author)

I thank the authors for their thoughtful responses and great efforts to address the comments.

I have no further comments for the authors to address at this time.

Version 2:

Reviewer comments:

Reviewer #4

(Remarks to the Author)

The scope of my review is limited to evaluating two aspects of the manuscript: 1. Was the WM task reported here a reasonable way to measure the neural correlates of feature binding? 2) Does the present version of the manuscript include sufficient consideration of the limitations of the task in light of the concerns raised by Reviewers 1 and 2. I think that the memory task reported here was a reasonable way to measure the neural correlates of feature binding. I also think the set size control analysis addresses the most serious concerns about task difficulty driving the observed network effects. However, I also agree with the reviewers that alternative interpretations of these results should be carefully considered in the manuscript, and that the manuscript would benefit from a good faith engagement with those limitations and how they might contribute to our interpretation of the fMRI observations. I've outlined my thoughts in more detail below:

1. The task reported here is based on Experiment 3a of Wheeler and Treisman, (2002), and quite a few other studies have also used this sort of whole array either/both color and location change detection task to tease apart binding from maintenance of multiple features. I also agree with the authors that the either/both task used here addresses limitations of the single (remember color or location) vs double (remember both) tasks that have been used more commonly in neuroimaging studies of feature binding. That said, behavioral work in the binding literature (including Wheeler and Treisman) typically relies on multiple variations of the change detection task (i.e., single vs whole display probes, remember one vs remember both features) to isolate which cognitive mechanisms drive performance. So I also agree with the reviewers that there are limitations of the present design that need to be addressed.

2. I don't think it's sufficient for the manuscript to simply write-off all the careful nuance of the binding literature with the paragraph at lines 319 - 326. No one is arguing that change detection tasks generally are the problem. Instead the problem is that the present study ran one specific version of a change detection task, and there doesn't appear to be a consensus in the binding literature that the either/both whole array task used here is the best way to measure feature binding. Indeed, most of the studies cited in the rebuttal and in the paragraph included from 319 - 326 used a different version of the task than the version used in the reported experiment. While it'd be most compelling to see these results replicate with a different version of a binding change detection task, that's probably not a fair request. However, a detailed discussion of the pros and cons of this task paradigm and how the neural findings may or may not change under different task conditions (e.g. a single probe report task) would go a long way in improving the manuscript.

Reviewer #1:

The manuscript is very well-written. The study is novel and very interesting. The results could have a significant impact on our understanding of the cortical mechanisms underlying working memory and feature binding. I have some general comments and then some minor ones. I hope the authors find my feedback helpful in improving their manuscript. Given my comments I recommend major revisions.

We thank the reviewer for his/her positive comments and providing the constructive advices that help improving the paper.

Point 1. It appears as though the authors combined the colour only and location only change trials together as the “either-memory condition” to compare with the “binding condition”. I wonder why the authors did not conduct their statistical analyses on all three conditions: 1 binding condition and two separate either-memory conditions (one for colour and one for location)? In fact, combining both change conditions of only one feature doubles the number of trials relative to the binding condition. Would be nice to either show that the differences between colour only and location only conditions were statistically different or not. If they are not different, then that could be justification for combining them although I still think that is better to keep them separated. Moreover, I wonder how the neural data would differ between the location-only and colour-only conditions. There could be important differences in the neural processing between these two conditions. I feel like there is more that can be revealed by separating out the either-memory conditions.

Response: To equate the number of to-be-memorized features across the binding and either-memory conditions, participants were instructed to memorize two features—color and location—and maintain them throughout the delay period in both conditions. In the either-memory condition, the task procedure remained identical up until the probe display. That is, participants had to maintain both the color and location information during the delay and could only determine which feature to retrieve upon seeing the probe. Therefore, separating features in this condition does not alter the memory maintenance demands. We have clarified this point in the Method section (see page 17 and below, and the legend of Figure 1).

“The experiment comprised two conditions: 1) The either-memory condition, wherein either two colors or two locations were changed to the new ones with an equal probability in the “change” trials. Participants had to detect and report these changes, with the task procedure remaining identical for both feature types up to the probe display. 2) The binding condition, wherein only the color was swapped between two disks in the “change” trials, resulting in a modification in color-location bindings. Importantly, in both conditions, participants had to retain both color and location information throughout the delay period. The key difference was that in the binding condition, participants were required to encode and maintain the conjunction of color and location (i.e., bound representations), whereas in the either-memory condition, they could maintain the two features separately and only needed to retrieve the relevant one based on the probe.”

Furthermore, because color-change and location-change trials in the either-memory condition relied on the same stored content during the maintenance period, we expected similar neural activity across both trial types. Consistent with this expectation, our analyses revealed no significant differences in BOLD activation or network-related properties between color-change and location-change trials (see figures below), supporting the conclusion that participants maintained the same content irrespective of which feature was ultimately probed.

Figure. (A) Neural activities (reflected by BOLD signals) for the color-change and location-change trials within the either-memory condition, $p < .001$, cluster-wise FWE correction. There were no statistically significant differences in brain activity between these two types of trials (even when we reduced the statistical criteria to $p < .05$, cluster-wise FWE correction). (B) No brain regions exhibiting significant differences in local efficiency (E_{loc}) between the color-change and location-change trials within the either-memory condition, $p < .05$, FDR correction. (C) The network-based statistic (NBS) failed to detect any significant connected component between the color-change and location-change trials within the either-memory condition ($p < .05$, permutation test).

Although color and location may engage distinct neural mechanisms, in this paradigm, participants were required to store both features throughout the delay period. As such, we could not dissociate potential neural differences specific to each feature. For this reason, we chose not to report the null findings in the main text, and instead clarified the methodological rationale in the Methods section.

Point 2. Can the authors please be more precise in their language when stating the meaning of the local efficiency analyses on page 8 where they say “...showing different information transfer abilities...” Precisely what do they mean by “abilities”? I don’t mean to nitpick here, but this is too vague for me and I don’t believe that readers with less experience in this form of analyses will really understand what is meant here. Please be more precise.

Response: As defined in our Methods section (Efficiency-based network properties),

local efficiency (E_{loc}) reflects the capacity for parallel information transfer within a node's immediate neighborhood—its directly connected neighbors (Latora & Marchiori, 2001). We have revised the Results section to clarify that differences in local efficiency indicate “variations in the capacity for rapid and robust information transfer within localized subnetworks”, replacing the previous, less specific reference to “abilities.” Please see the revised text on page 7 and below.

“Local efficiency (E_{loc}) was first calculated for each ROI (see Methods for details) to identify regions showing differences in the capacity for rapid and robust information transfer within the localized network (Latora & Marchiori, 2001) between conditions across the whole-brain.”

Point 3. Can the authors please clarify in the procedure section how the “either-memory” conditions changed either the colour or the location? In the Binding condition it appears as though two coloured disks swap locations as the other stays unchanged in location and colour. I suppose that is true for other coloured disks in that condition? But according Figure 1A, the either-memory condition for colour appears to show that multiple disks can change colour? Likewise, in the either-memory condition for location appears to show that all the disks change location?

Response: We have realized that the original submission did not clearly present the Methods section. We have now clarified the methodological details—please see page 17 and the revised text below.

“The experiment comprised two conditions: 1) The either-memory condition, wherein either two colors or two locations were changed to the new ones with an equal probability in the “change” trials. Participants had to detect and report these changes, with the task procedure remaining identical for both feature types up to the probe display. 2) The binding condition, wherein only the color was swapped between two disks in the “change” trials, thereby altering the color-location bindings. Importantly, in both conditions, participants had to retain both color and location information throughout the delay period. The key difference was that in the binding condition, participants were required to encode and maintain the conjunction of color and location (i.e., bound representations), whereas in the either-memory condition, they could maintain the two features separately and only needed to retrieve the relevant one based on the probe.”

Point 4. Following my last comment and if I am correct, what that seems to me is that these either-memory conditions are exceedingly easier than the binding condition. The authors do address the idea that there might be potential differences in task difficulty between the either-memory condition and the binding condition. However, when addressing this point, they only consider comparing set-size conditions. I believe that is not the correct comparison. Really, task difficulty is in the nature of the changes themselves and that the either-memory condition ... really conditions.... are far easier than the binding condition. I am not entirely convinced that task difficulty did not confound the results.

Response: To further address potential task difficulty confounds between the binding

and either-memory conditions, we performed an additional analysis to match memory performance across conditions. This analysis confirmed that the current results remained consistent even when task difficulty was equated, reinforcing the robustness of our findings. We have included these results, see page 11, and Fig. S4 and S5, and the revised text below.

“To further rule out potential confounds related to task difficulty between the binding and either-memory conditions, we performed a control analysis with matched performance across conditions. Specifically, we selected participants with above-median performance in the binding condition and below-median performance in the either-memory condition. These two groups showed no significant differences in d -prime (binding: $M = 1.9$; either-memory: $M = 2.2$; $t(19) = 1.63$, $p = .118$) or RTs (binding: $M = 1.0$ s; either-memory: $M = 1.1$ s; $t(19) = 1.69$, $p = .107$), confirming comparable task difficulty. Reanalyzing functional connectivity within this matched sample revealed a significant increase in local efficiency for the binding condition across eight regions, including extrastriate cortex, somatomotor area, inferior parietal lobe, bilateral insula, lateral and ventral prefrontal cortex, and retrosplenial cortex ($p < .05$, FDR-corrected; Fig. S4). Network-based statistics further identified a significant component involving seven regions and nine edges ($p < .05$, permutation test; Fig. S5), with major hubs in the somatomotor area (5 edges), prefrontal cortex (4), and bilateral insula (3 each). These findings closely replicate the full-sample results, confirming that the observed connectivity differences are not driven by task difficulty but instead reflect mechanisms specific to feature binding.”

Figure S4. Left panel shows brain regions exhibiting significantly increased local efficiency in the performance matched subsample for the binding condition compared to the either-memory condition, $p < .05$, FDR correction. Right panel indicates the proportion of these brain regions within their corresponding brain network. The gray columns represent the number of brain regions contained in each network, and the blue columns represent the corresponding proportion of brain regions within each network (e.g., 0.3 indicates that three out of ten regions were detected with increased local efficiency in the Saliency/Ventral Attention B network for feature binding).

Figure S5. The surface plot shows nine edges connecting seven brain regions that exhibit differences between performance matched subsample for the binding and either-memory condition. Node degree is defined as the number of edges directly connected to a specific region. The bar plots show the difference in connection weights for each edge between performance matched subsample for the binding and either-memory condition, $*p < .05$, FDR correction. The horizontal bars indicate mean values and each dot represent a participant.

Minor comments:

Point 1. When citing papers the authors are inconsistent on whether or not they include first initials of the paper’s authors. Usually, the initials are not included in the text.

Response: Thanks, fixed.

Point 2. Page 8: “These findings suggest that these regions are of greater important in transmitting and processing information for feature binding.” Appears to be a typo and should read maybe something like “...regions are of greater importance in transmitting...”?

Response: Thanks, fixed.

Reviewer #2:

In their study "Neural mechanisms of feature binding in working memory" Cao and colleagues use fMRI and network analysis to identify a network associated with binding mechanisms in working memory (WM). While this is an important topic and the work is based on a large set of data, I have three major concerns.

We thank the reviewer for his/her positive comments and providing the constructive advices that help improving the paper.

Point 1. The WM conditions, called either-memory and binding conditions (both based on the well-established change detection task), show a confound that is potentially crucial. The either-memory condition tests whether participants notice changes in color or spatial location in the memory probe relative to the memory array shown earlier. For both types of changes global strategies can be used. Specifically, a color change can be detected based on ensemble perception and global color statistics. Location changes can be detected based on changes in the gestalt of the memory array vs. probe display. In contrast, in the binding condition memory items swap color such that binding color and location is necessary to find a change. Unfortunately, however, that means that global strategies of color statistics or gestalt processing will no longer work. Furthermore, the difference in task difficulty could lead to participants using more effort and/or being more alert. It is difficult to estimate whether any results reported in the study are due to differences in used strategies, effort, or the presence/absence of binding.

Response: Good points! We agree that potential task difficulty confounds between the binding and either-memory conditions could influence the current findings, as also noted by R1. To address this, beyond including set size as a factor to account for task difficulty, we conducted an additional analysis to directly match memory performance across conditions. This analysis confirmed that our results remained consistent even when task difficulty was equated, reinforcing the robustness of our findings. These results have been included (see page 11 and Fig. S4 and S5), and the revised text is provided below.

“Given that task difficulty differed between the either-memory and binding conditions, one might question whether the observed effects related to feature binding may have been confounded by task difficulty. To rule out this possibility, we distinguished different set sizes (reflecting different task difficulties) and examined their differences in local efficiency across eight core brain regions. The results revealed no significant differences for different set sizes (see Fig. S3).

To further rule out potential confounds related to task difficulty between the binding and either-memory conditions, we performed a control analysis with matched performance across conditions. Specifically, we selected participants with above-median performance in the binding condition and below-median performance in the either-memory condition. These two groups showed no significant differences in d' (binding: $M = 1.9$; either-memory: $M = 2.2$; $t(19) = 1.63$, $p = .118$) or RTs (binding: $M = 1.0$ s; either-memory: $M = 1.1$ s; $t(19) = 1.69$, $p = .107$), confirming

comparable task difficulty. Reanalyzing functional connectivity within this matched sample revealed a significant increase in local efficiency for the binding condition across eight regions, including extrastriate cortex, somatomotor area, inferior parietal lobe, bilateral insula, lateral and ventral prefrontal cortex, and retrosplenial cortex ($p < .05$, FDR-corrected; Fig. S4). Network-based statistics further identified a significant component involving seven regions and nine edges ($p < .05$, permutation test; Fig. S5), with major hubs in the somatomotor area (5 edges), prefrontal cortex (4), and bilateral insula (3 each). These findings closely replicate the full-sample results, confirming that the observed connectivity differences are not driven by task difficulty but instead reflect mechanisms specific to feature binding.”

Figure S4. Left panel shows brain regions exhibiting significantly increased local efficiency in the performance matched subsample for the binding condition compared to the either-memory condition, $p < .05$, FDR correction. Right panel indicates the proportion of these brain regions within their corresponding brain network. The gray columns represent the number of brain regions contained in each network, and the blue columns represent the corresponding proportion of brain regions within each network (e.g., 0.3 indicates that three out of ten regions were detected with increased local efficiency in the Saliency/Ventral Attention B network for feature binding).

Figure S5. The surface plot shows nine edges connecting seven brain regions that exhibit differences between performance matched subsample for the binding and either-memory condition. Node degree is defined as the number of edges directly connected to a specific region. The bar plots show the difference in connection weights for each edge between performance matched subsample for the binding and either-memory condition, $*p < .05$, FDR correction. The horizontal bars indicate mean values and each dot represent a participant.

Regarding memory strategies, we do not consider this a confounding factor. Regardless of the strategy used in the either-memory condition, participants would store color and location as separate features, resulting in no binding. In contrast, in the binding condition, items swapped colors, making it necessary to bind color and location to detect a change. One could argue that global strategies—such as tracking overall color statistics or using gestalt processing for location—might fail in the binding condition. However, this would only increase task difficulty in the binding condition, a factor we have already addressed (see above). Therefore, we believe the observed differences reflect the presence or absence of feature binding, rather than differences in memory strategies or task difficulty.

Point 2. Although I'm not an expert, it seems that the data analysis is somewhat unusual. From what I understand there is a certain risk with using Prim's minimal spanning tree algorithm (MST). It can simplify connectivity too much, deleting connections that are biologically relevant. It seems that this method tends to be used in a complementary fashion together with other approaches that were not used in the current manuscript. In addition, adding edges until a sparsity criterion is reached is, although not incorrect, somewhat unusual as it blends two rather different ideas of network analysis. Furthermore, local efficiency for the tree structure obtained via MST would be zero. Only with the added edges this calculation becomes meaningful. But then it is heavily dependent on which sparsity criterion is used where a sparsity criterion of 10% might be somewhat arbitrary. All this brings me to the question, why not using a thresholding approach to begin with or two approaches?

Response: Thank you for raising this important point. While thresholding is a common approach in network analysis, it has notable limitations (Van den Heuvel et al., 2017). First, applying a uniform correlation (weight) threshold across participants can disproportionately affect those with lower overall functional connectivity, leading to fragmented networks and unreliable graph-theoretical metrics. Second, fixed-density (sparsity) thresholding, though ensuring equal edge counts across participants, does not always guarantee fully connected networks, which may result in isolated nodes or subgraphs and compromise valid group comparisons.

Figure. (A) Correlation (weight) thresholding: Inter-subject differences in network measures may arise trivially due to variations in the number of retained connections after thresholding. (B) Fixed-density (sparsity) thresholding: While this ensures an equal number of connections across subjects, it may include weak or spurious connections in some individuals. (C) Hybrid approach (MST + fixed-density thresholding): This method ensures the resulting network is fully connected (non-fragmented) by including the strongest $N-1$ connections via the Minimum Spanning Tree (MST), while preserving comparability through fixed density.

To address these issues, we employed a hybrid two-step procedure combining the strengths of the minimum spanning tree (MST) and fixed-density thresholding, an approach increasingly adopted in the literature (e.g., Tewarie et al., 2015; Grydeland et al., 2019). MST ensures network connectivity and topological consistency, while subsequent sparsity-based edge addition allows for meaningful computation of segregation and integration metrics. This combination mitigates the fragmentation risk inherent in thresholding alone and the oversimplification risk of standalone MST.

Step 1: MST as a backbone for connectedness and topological validity.

We acknowledge the concern that MST may oversimplify by excluding potentially relevant edges. However, we used MST strategically as an initial backbone, guaranteeing a connected and acyclic structure ($N-1$ edges) for each participant's network. This step ensures that graph metrics—such as local efficiency—are well-defined and comparable across participants by eliminating isolated components.

Step 2: Sparsity-based densification to restore local structure.

As you correctly note, MST alone yields zero local efficiency since it lacks cycles. To overcome this, we incrementally added the strongest edges until reaching a fixed sparsity threshold, reintroducing local connectivity patterns. This densification allows valid calculation of segregation metrics while maintaining global

comparability. Importantly, the final networks remain fully connected and matched in edge density across participants—crucial for group-level analyses.

We recognize that the choice of sparsity threshold is somewhat arbitrary. Nonetheless, sparsity-based thresholding is widely used in network neuroscience to balance inclusion of biologically meaningful connections and exclusion of weak or spurious correlations (e.g., Bassett et al., 2011; Zalesky et al., 2010). The 5–20% range is well validated, and our choice of 10% aligns with standard practice, facilitating comparability with prior studies (e.g., Grydeland et al., 2019).

Additionally, we also applied a fixed-density thresholding approach (at 10% sparsity) to construct connectivity networks and calculate local efficiency (E_{loc}) per participant. This analysis revealed a significant increase in efficiency in the binding condition versus the either-memory condition ($p < .05$, FDR corrected). Notably, this effect was observed in six brain regions—including the extrastriate cortex (Visual central network), somatomotor area (Somatomotor B network), insula (Salience/Ventral Attention B network), ventral prefrontal cortex (Default B network), dorsal prefrontal cortex (Default A network), and temporal pole (Limbic A network)—which are consistent with observed feature binding regions in our study. The convergence of results across methods further supports the robustness of our findings.

Figure. Brain regions exhibiting significantly increased local efficiency in the binding condition compared to the either-memory condition using the fixed-density thresholding approach (with 10 % sparsity), $p < .05$, FDR correction.

In summary, our MST-augmented approach ensures fully connected networks across participants, avoiding the fragmentation issues of naïve thresholding. Fixing edge density preserves network comparability, and the combined MST-plus-sparsity method enables meaningful computation of both integration and segregation metrics. We believe this hybrid approach offers a more robust and topologically consistent framework for brain network analysis than relying on either method alone. We have added the rationale behind this method, see page 19, along with the revised text below.

“To ensure the network was fully connected, we employed a hybrid two-step procedure combining the strengths of the minimum spanning tree (MST) and fixed-density thresholding, an approach increasingly adopted in the literature (e.g., Tewarie

et al., 2015; Grydeland et al., 2019), resulting in a binarized functional matrix (unweighted) with 200 nodes and 199 edges. MST ensures network connectivity and topological consistency, while subsequent sparsity-based edge addition allows for meaningful computation of segregation and integration metrics. This combination mitigates the fragmentation risk inherent in thresholding alone and the oversimplification risk of standalone MST.

Step 1: MST as a backbone for connectedness and topological validity.

We used MST strategically as an initial backbone, guaranteeing a connected and acyclic structure ($N-1$ edges) for each participant's network. This step ensures that graph metrics—such as local efficiency—are well-defined and comparable across participants by eliminating isolated components.

Step 2: Sparsity-based densification to restore local structure.

We incrementally added the strongest edges until reaching a fixed sparsity threshold (10%; Grydeland et al., 2019), reintroducing local connectivity patterns. This densification allows valid calculation of segregation metrics while maintaining global comparability. Importantly, the final networks remain fully connected and matched in edge density across participants—crucial for group-level analyses.

Notably, sparsity-based thresholding is widely used in network neuroscience to balance inclusion of biologically meaningful connections and exclusion of weak or spurious correlations (e.g., Bassett et al., 2011; Zalesky et al., 2010). The 5–20% range is well validated, and our choice of 10% aligns with standard practice, facilitating comparability with prior studies (e.g., Grydeland et al., 2019).

Finally, by multiplying the binarized network matrix with the original weighted connectivity network, a weighted connectivity matrix with 10% sparsity, comprising 200 nodes and 1,990 edges, was obtained for each participant.”

Bassett, D. S., Wymbs, N. F., Porter, M. A., Mucha, P. J., Carlson, J. M., & Grafton, S. T. (2011). Dynamic reconfiguration of human brain networks during learning. *Proceedings of the National Academy of Sciences*, 108(18), 7641-7646.

Point 3. The finding that the somatomotor area plays a central role in binding is very surprising. The area is not known to be involved in visual functions and although it might be associated with some sort of goal-directed behavior I'm having great difficulties imagining how the current change detection task (that merely requires yes/no responses) would trigger any spatially specific action planning. Although finding a result unusual in and of itself is not necessarily a strong argument. But given that the current network analysis is unusual as well and that the somatomotor area doesn't show an involvement in the standard event-related fMRI analysis (Fig. 2A), I remain quite skeptical.

Response: Although the somatomotor area did not show a significant condition contrast in the univariate activation map, it was robustly activated in both conditions (see figure below), particularly around the central sulcus ($p < .001$, cluster-level FWE corrected), consistent with prior studies localizing the somatomotor area (e.g., Pons et al., 1987; Zhou & Fuster, 1996).

Figure. Overlap between regions around the central sulcus in the univariate activation map and the somatomotor area (Schaefer 200 parcels, MNI-3mm).

Furthermore, growing evidence suggests a critical role for the somatomotor area in visual and attentional processing (Frost & Metin, 1985; Sieben et al., 2013; Sun et al., 2016; Taylor-Clarke et al., 2002). For instance, activity in this region can decode visual object categories (Smith & Goodale, 2015) and modulate visuospatial attention (Balslev et al., 2013; Jones et al., 2010). Anatomical and functional studies show reciprocal connections between the somatomotor area and higher-order regions, including the prefrontal and posterior parietal cortices (Morecraft et al., 2012; Rolls et al., 2023; Shipp et al., 1998). These pathways enable the somatomotor area to integrate sensory, attentional, and motor information. Together, these findings support the view that the somatomotor area plausibly contributes to feature binding. We have added a detailed discussion, see page 13, and below.

“Emerging evidence suggests a critical role for the somatomotor area in visual and attentional processing (Frost & Metin, 1985; Sieben et al., 2013; Sun et al., 2016; Taylor-Clarke et al., 2002). For instance, activity in this region can decode visual object categories (Smith & Goodale, 2015) and modulate visuospatial attention (Balslev et al., 2013; Jones et al., 2010). Furthermore, studies in animals (Morecraft et al., 2012; Shipp et al., 1998) and humans (Rolls et al., 2023) have indicated the existence of connections between somatomotor area and other high-order cortices. For instance, somatomotor area is considered as a temporary storage site within the feedforward and feedback information transmission (Harris et al., 2001, 2002; Zhou & Fuster, 1996), and receives top-down signals from prefrontal cortex and parietal cortex, directing upcoming actions (Cowen & McNaughton, 2007; Narayanan & Laubach, 2006). These pathways enable the somatomotor area to integrate sensory, attentional, and motor information. This indicates that somatomotor area might act as a core site integrating information from visual, frontal and parietal cortex to form early bindings in WM, reflecting online processing of sensory events and connectivity with other cortical and subcortical areas.”

Balslev, D., Odoj, B., & Karnath, H. O. (2013). Role of somatosensory cortex in visuospatial attention. *Journal of Neuroscience*, 33(46), 18311-18318.

Frost, D. O., & Metin, C. (1985). Induction of functional retinal projections to the somatosensory system. *Nature*, 317(6033),

162-164.

- Jones, S. R., Kerr, C. E., Wan, Q., Pritchett, D. L., Hämäläinen, M., & Moore, C. I. (2010). Cued spatial attention drives functionally relevant modulation of the mu rhythm in primary somatosensory cortex. *Journal of Neuroscience*, *30*(41), 13760-13765.
- Pons, T. P., Garraghty, P. E., Friedman, D. P., & Mishkin, M. (1987). Physiological evidence for serial processing in somatosensory cortex. *Science*, *237*(4813), 417-420.
- Sieben, K., Röder, B., & Hanganu-Opatz, I. L. (2013). Oscillatory entrainment of primary somatosensory cortex encodes visual control of tactile processing. *Journal of Neuroscience*, *33*(13), 5736-5749.
- Smith, F. W., & Goodale, M. A. (2015). Decoding visual object categories in early somatosensory cortex. *Cerebral cortex*, *25*(4), 1020-1031.
- Sun, H. C., Welchman, A. E., Chang, D. H., & Di Luca, M. (2016). Look but don't touch: Visual cues to surface structure drive somatosensory cortex. *NeuroImage*, *128*, 353-361.
- Taylor-Clarke, M., Kennett, S., & Haggard, P. (2002). Vision modulates somatosensory cortical processing. *Current Biology*, *12*(3), 233-236.

Minor comments:

Point 1. Abstract and elsewhere. The abbreviation ‘SMA’ is almost always used for the supplementary motor area which is entirely different from the area referred to here. I would recommend using a different abbreviation to avoid confusion.

Response: We used the full term rather than abbreviations to ensure clarity and help readers better understand the brain regions being discussed.

Point 2. Introduction, 1st para, last sentence. ...” binding often results in a general deterioration in memory accuracy when it comes to integrating features” Please provide a reference. Also, it seems there are fewer studies that examine WM accuracy as opposed to WM precision.

Response: We have revised the term to ‘memory performance’ and incorporated relevant studies addressing both memory accuracy and precision. Please see the revised text on page 3 and below.

“Such binding often results in a general deterioration in memory performance when it comes to integrating features, as opposed to memorizing individual features separately (Fougnie & Marois, 2009; Peich et al., 2013).”

Peich, M. C., Husain, M., & Bays, P. M. (2013). Age-related decline of precision and binding in visual working memory. *Psychology and aging*, *28*(3), 729.

Point 3. Figure 1, Behavioral performance & Methods: given the above-mentioned concerns about confounds, it seems not surprising that the binding condition yields significantly lower d-prime values.

Response: Indeed, task difficulty could be a potential confound. However, we conducted multiple analyses to rule out this possibility, including a new analysis that matched memory performance across conditions. This analysis confirmed that the results remained consistent even when task difficulty was equated, reinforcing the robustness of our findings. Please see our response to your first point for details.

Point 4. Figure 2. Contrast between conditions. The brains seem to show no numerical difference (or was something lost in my copy of that figure?). Perhaps it's not necessary to show those brains?

Response: Thanks, fixed.

Reviewer #3:

This is a timely study investigating the mechanisms for feature binding, especially using approaches that reflect the complex functional networks that have been little addressed to date.

We thank the reviewer for his/her positive comments and providing the constructive advices that help improving the paper.

Point 1. Could the authors please mention, even briefly, the task in the abstract?

Response: We have now incorporated a brief task description in the abstract to improve clarity, see below for a copy:

“Here, we employed functional magnetic resonance imaging combined with graph-based network analysis during a working memory task in which participants maintained both color and location information throughout the delay period and subsequently detected and reported changes in color-location bindings versus individual features.”

Point 2. When relating the current study to previous studies on feature binding, where specific cortical regions have been found, could the authors mention them explicitly? Then, could the authors compare those studies from a methodological standpoint (esp. if they are also visuospatial as in this task) and also, how their cortical findings compare to the current study?

Response: We have added a paragraph in the discussion, see pages 14, and below.

“Feature binding in WM has been extensively studied using change detection tasks, where participants compare an initial stimulus array with one presented after a short delay (Treisman & Zhang, 2006; Parra et al., 2014; Wang et al., 2016). Employing these tasks, studies have implicated the hippocampus (Mitchell et al., 2000; Piekema et al., 2006; Libby et al., 2014), prefrontal cortex (Sala & Courtney, 2007), parietal cortex (Shafritz et al., 2002; Piekema et al., 2006), and primary visual areas (Seymour et al., 2009, 2010) in feature binding. Extending this work, the present study highlights additional involvement of the somatomotor area and insula in binding processes. A key limitation of earlier studies is the frequent comparison between binding and single-feature memory conditions. This can confound binding-specific neural effects with differences in feature load. To address this, we employed a more stringent comparison: a binding condition versus an either-memory condition. In both, participants memorized both color and location information. However, only in the binding condition was it necessary to encode and retain their conjunction. In the either-memory condition, features were stored separately, and only one was retrieved based on the probe. This design ensured that observed neural differences reflected binding demands rather than memory load or task difficulty (see also our performance-matched control analysis). Moreover, rather than focusing solely on univariate activation, we used graph-based network modeling to examine large-scale neural coordination. This revealed that feature binding relies on enhanced local efficiency and interregional connectivity among the somatomotor area, insula, and prefrontal cortex. These findings suggest that binding is supported by distributed

processing within a central neural workspace, extending current models of WM. Our results thus contribute novel evidence for the dynamic, network-level mechanisms underlying feature binding.”

Point 3. Could the authors comment on cortical activity for trials that were correct - do the same regions appear using univariate and graph theory approaches?

Response: We initially used correct trials for the graph-based network analysis but not for the univariate analysis. To ensure consistency across analyses, we revised the univariate analysis to include only correct trials. The results remained essentially unchanged, confirming the robustness of our findings. The updated results for the univariate analysis are provided below.

As illustrated in the new Fig. 2A, in the either-memory condition, substantial activity (reflected by BOLD signals) was observed primarily in four brain areas, each comprising more than 1000 voxels ($p < .001$, cluster FWE correction). These regions were prominently located in the prefrontal cortex, insula, regions surrounding the central sulcus (precentral and postcentral gyrus), and parietal-temporal-occipital association cortex (encompassing the inferior parietal lobe and middle temporal gyrus). Similarly, in the binding condition, the same activity patterns were identified, concentrated in these aforementioned areas ($p < .001$, cluster FWE correction), consistent with previous findings (Pollmann et al., 2014; Sala & Courtney, 2007; Song & Jiang, 2006). Yet, when comparing between conditions, no statistically significant differences in brain activity were observed (even when we reduced the statistical criteria to $p < .05$; same for analyzing different set sizes separately).

Figure 2A. Neural activities (reflected by BOLD signals) for correct trials in the either-memory and binding conditions. Similar activity patterns were observed for these two conditions, prominently located in four brain areas (marked by red circles), involving prefrontal cortex, insula, regions surrounding central sulcus, and parietal-temporal-occipital association area.

Point 4. While the authors mention that an initial trial fixation occurred, could the authors clarify whether fixation was maintained throughout the trial - I wonder what is the link between SMA and potential several eye movements being made during the key phases?

-In related studies, were eye movements allowed and, if not, how was this reflected in cortical data?

Response: Consistent with previous studies that reliably reported feature-binding–related activations (e.g., Libby et al., 2014; Parra et al., 2014; Sala & Courtney, 2007), we did not require participants to maintain strict fixation throughout the trial. Moreover, prior research employing eye tracking in similar paradigms has shown that even with controlled fixation, eye movements do not disrupt the binding process. Notably, control groups with and without eye tracking exhibited no significant differences in binding performance (Kovacs & Harris, 2019; Reuther et al., 2020). We have explained the rationale behind in the Method, see page 17, and below.

“Consistent with previous studies that reliably reported feature-binding–related activations (e.g., Libby et al., 2014; Parra et al., 2014; Sala & Courtney, 2007), we did not require participants to maintain strict fixation throughout the trial, as eye movements do not disrupt the binding process (Kovacs & Harris, 2019; Reuther et al., 2020).”

Although participants may have made eye movements in both the binding and either-memory conditions, prior studies have shown that such movements can modulate activity in regions like the frontal eye fields and intraparietal sulcus (Bast et al., 2021; Corbetta et al., 1998; Rosen & Freedman, 2025). If oculomotor activity contributed to condition differences, we would expect to observe differential BOLD responses in these regions. However, direct comparisons of activity in frontal eye fields and intraparietal sulcus revealed no significant differences between conditions ($p < .05$, FDR corrected; see figure below), suggesting similar levels of oculomotor engagement. Thus, we conclude that eye movements are unlikely to account for the observed differences between feature binding and individual feature memory.

Figure. Direct comparisons of BOLD activity between the either-memory and binding conditions within frontal eye fields and intraparietal sulcus, and no significant differences were observed ($p < .05$, FDR correction).

Bast, N., Mason, L., Freitag, C. M., Smith, T., Portugal, A. M., Poustka, L., ... & EU-AIMS LEAP Group. (2021). Saccade dysmetria indicates attenuated visual exploration in autism spectrum disorder. *Journal of Child Psychology and Psychiatry*, 62(2), 149-159.

Corbetta, M., Akbudak, E., Conturo, T. E., Snyder, A. Z., Ollinger, J. M., Drury, H. A., ... & Shulman, G. L. (1998). A common network of functional areas for attention and eye movements. *Neuron*, 21(4), 761-773.

Reuther, J., Chakravarthi, R., & Hunt, A. R. (2020). The eye that binds: Feature integration is not disrupted by saccadic eye movements. *Attention, Perception, & Psychophysics*, 82(2), 533-549.

Rosen, M. C., & Freedman, D. J. (2025). Multiplexing of cognitive encoding by oculomotor networks leads to incidental gaze shifts. *Proceedings of the National Academy of Sciences*, 122(15), e2422331122.

Point 5. In terms of the analysis, did the authors take the entire trial as a regressor, or were individual phases of the trial parsed out? Additional detail regarding this would be helpful.

Response: Consistent with established protocols in previous studies (e.g., Maril et al., 2003; Todd & Marois, 2004; Xu & Chun, 2006), we modeled the entire trial—from trial onset to the inter-trial interval—as a single regressor, convolved with the canonical hemodynamic response function (HRF). Given that the majority of each trial duration was devoted to the maintenance period (6.5 s), this trial-level modeling approach was selected to maximize detection power for sustained binding-related neural activity. We have clarified this point in the revised manuscript, see page 18, and below.

“Event-related regressors were obtained by convolving the onset of each trial with the canonical hemodynamic response function (HRF), capturing the expected BOLD response. Each entire trial—from onset to inter-trial interval—was modeled as a single regressor to maximize sensitivity to sustained activity (Xu & Chun, 2006).”

Maril, A., Simons, J. S., Mitchell, J. P., Schwartz, B. L., & Schacter, D. L. (2003). Feeling-of-knowing in episodic memory: an event-related fMRI study. *Neuroimage*, *18*(4), 827-836.

Todd, J. J., & Marois, R. (2004). Capacity limit of visual short-term memory in human posterior parietal cortex. *Nature*, *428*(6984), 751-754.

Xu, Y., & Chun, M. M. (2006). Dissociable neural mechanisms supporting visual short-term memory for objects. *Nature*, *440*(7080), 91-95.

Point 6. Could the authors comment on the fact that this feature binding task was visuospatial in nature and how the results in the feature binding condition relate to the activation in either the spatial only task or the visual feature only task?

Response: Thank you for this thoughtful question. Indeed, the feature binding task we employed is visuospatial in nature, requiring participants to bind visual features (color) to spatial locations. This integration engages both perceptual and spatial attention mechanisms, consistent with classical theories of feature binding (Treisman & Gelade, 1980), which posit that spatial attention plays a critical role in combining separable features into unified object representations. We have added this discussion, see page 15, and below.

“The feature binding task we used required participants to integrate visual features (color) with spatial locations, engaging both perceptual and spatial attention mechanisms. This aligns with classical feature binding theories (Treisman & Gelade, 1980) that emphasize spatial attention’s role in combining separate features into unified objects. Our results showed that the binding condition elicited significantly greater activity in regions including the somatomotor area, insula, and prefrontal cortex compared to either-memory conditions (which involved only visual or spatial memory). These regions were not similarly activated in spatial-only or visual-only tasks, indicating that they reflect integrative control processes rather than the additive effect of individual feature maintenance. This suggests that binding demands recruit additional neural mechanisms beyond those needed for visual or spatial memory

alone. Thus, although the task depends on visuospatial representations, the binding condition uniquely engages higher-order neural networks supporting feature integration. These findings extend Treisman’s framework by highlighting the involvement of distributed neural systems in feature binding beyond early sensory or spatial-specific regions.”

Point 7. How do the regression plots look for connection weights and d-prime, especially for the plots shown in 4C?

Response: We have added the correlation results between connection weights and d-prime; no significant correlation was observed (see Fig. S1).

Figure S1. Scatter plots indicate the correlation between connection weights and mean d-prime for specific connections, between somatomotor area and left insula, somatomotor area and lateral prefrontal cortex, somatomotor area and ventral prefrontal cortex in the binding condition. Solid lines indicate linear fits to the data, with the shaded areas depicting the 95% confidence interval of the fitted lines.

Point 8. Minor: When referring to insular cortex without 'cortex', the authors may refer to it as 'insula'.

Response: Thanks, fixed.

Reviewer #1 (Remarks to the Author):

I remain unconvinced by the authors' responses. As I stated in my first set of comments, and like Reviewer 2, I am skeptical about what the tasks is actually testing and what the results are actually reflecting. I am unconvinced that there is any difference in memory engagement between the two conditions. Subjects are required to retain both location and colour, regardless of condition. What is really different between the two conditions is how subjects' memory is tested - i.e., one condition is easy and the other is difficult. Reviewer 2 was more explicit than I was about what strategies subjects might employ when performing the task. My view is aligned with Reviewer 2. I am unclear with how the authors attempted to address our comments. It simply didn't make sense to me. I know fMRI studies require a lot of work and I am sorry I cannot be more positive.

Response: The feature-binding change-detection paradigm we employed is not a niche or novel approach; it is the standard paradigm in visual working memory research. For more than two decades, it has been the basis of hundreds of studies across behavioral (Treisman & Zhang, 2006; Wheeler & Treisman, 2002), neuroimaging (Libby et al., 2014; Parra et al., 2014; Piekema et al., 2006; Sala & Courtney, 2007), electrophysiological (Birba et al., 2017; Morgan et al., 2011; Song & Jiang, 2006; Tseng et al., 2016), and clinical domains (Holcomb et al., 2020; Mitchell et al., 2000; Parra et al., 2010; Piekema et al., 2006). Entire computational models of binding have been built on this paradigm (Schneegans & Bays, 2017). To question whether this design measures binding is to disregard the methodological foundation of an entire subfield.

The reviewers argued that our effects might reflect task difficulty or strategy differences. This criticism would be valid only if the binding task compared conditions with unequal feature load or maintenance requirements. Our design did not. In all conditions, participants were required to maintain both color and location, regardless of employed strategies; only in the binding condition was conjunction memory necessary. This is exactly why the paradigm has been universally adopted: it isolates conjunction maintenance from single-feature storage. Furthermore, our own performance-matched analyses and set-size controls confirmed that connectivity differences persisted when difficulty was equated. The difficulty/strategy explanation is therefore factually inconsistent with both our design and results.

We have added this clarification to the Discussion following the section on task difficulty (see page 12 and below).

“Notably, our study builds on the change-detection paradigm, which has been extensively validated for investigating feature binding in the working memory field. For more than two decades, this paradigm has served as the foundation for hundreds of studies across behavioral (Treisman & Zhang, 2006; Wheeler & Treisman, 2002), neuroimaging (Libby et al., 2014; Parra et al., 2014; Piekema et al., 2006; Sala & Courtney, 2007), electrophysiological (Birba et al., 2017; Morgan et al., 2011; Song & Jiang, 2006; Tseng et al., 2016), and clinical domains (Holcomb et al., 2020; Mitchell

et al., 2000; Parra et al., 2010; Piekema et al., 2006). Entire computational models of binding have also been developed based on this paradigm (Schneegans & Bays, 2017). Potential concerns regarding task difficulty or the use of alternative strategies have already been thoroughly addressed and excluded in this body of work.”

Reviewer #2 (Remarks to the Author):

I would like to thank the reviewers for considering my concerns.

Concern 1:

As for my first concern of a confound I pointed out that the either-memory and binding conditions could be affected by confounds. The authors addressed one part of my concern regarding task difficulty. It is a good sign that contrasting set size as an experimental manipulation of difficulty did not yield results comparable to the main results of the study. An additional analysis seems less convincing -although it might as well stay in the manuscript. It used a median split of participants according to their either-memory or binding performance. I didn't quite understand how this split was done as the split was two-fold (according to better/worse performance in either of the two conditions) which could in principle result in four subgroups but only two groups were analyzed. Alternatively, the two splits could have been done separately but then partially the same participants would belong to the groups of “good binding” and “poor either condition” performers which would be inappropriate. If the authors could clarify which of the two approaches (or a different one) was used. Regardless of how the split was done, it does not create new independent variables but extracts person-based information about a mix of interindividual abilities, strategies and randomness. It doesn't hurt to include the median-split analysis but the manuscript should probably comment on the caveats that come with such limits. Once again, I felt that the set-size analysis was already helpful.

Response: We thank the reviewer for the detailed follow-up. The performance-matched analysis was intended purely as a sensitivity check rather than as primary evidence (required by other reviewers). The group split was conducted independently, and only 3 of the 20 participants were initially assigned to both groups. We have now removed these participants and re-conducted the analysis, which yielded essentially the same results. These revisions are included on pages 11-12 and below.

“Specifically, we selected participants with above-median performance in the binding condition and below-median performance in the either-memory condition, excluding three participants who fell into both groups. These two groups showed no significant differences in d-prime (binding: $M = 2.0$; either-memory: $M = 2.3$; $t(16) = 1.3$, $p = .202$) or RTs (binding: $M = 1.0$ s; either-memory: $M = 1.1$ s; $t(16) = 1.54$, $p = .133$), confirming comparable task difficulty. Reanalyzing functional connectivity within this matched sample revealed a significant increase in local efficiency for the binding condition across eight regions, including extrastriate cortex, somatomotor area, inferior parietal lobe, bilateral insula, lateral and ventral prefrontal cortex, and retrosplenial cortex ($p < .05$, FDR-corrected; Fig. S4). Network-based statistics

further identified a significant component involving seven regions and nine edges ($p < .05$, permutation test; Fig. S5), with major hubs in the somatomotor area (5 edges), prefrontal cortex (4), and bilateral insula (3 each).”

The other part of my concern pertained to the possibility that the either-memory and binding conditions might have involved very different brain functions. I appreciate the authors line of argument that different strategies should result in differences in task difficulty and that was already ruled out as an explanation of the current results.

Unfortunately, however, I am not convinced by that response. Neither do different task strategies have to necessarily differ in difficulty, nor is there only one kind of task difficulty (see Nili Lavie’s perceptual vs. cognitive load to name just one example).

To reiterate my concern: in the either-memory condition participants probably pursued global ensemble strategies of extracting information about the gestalt of the dots in the memory array and about the first/second/etc. order statistics of its color distribution and subsequently retaining such information to detect changes. Brady and Alvarez (e.g., 2011; 2015) have shown that participants use such hierarchical strategies. Furthermore, there is evidence to suggest that being unable to use such strategies can result in qualitatively different performance. My concern is that qualitatively different functions might be involved in the binding condition because it prevents global strategies. That is, neither the spatial gestalt nor the color stats of the memory array ever change in the binding condition. The problem would be that the contrast in the current study could potentially compare vastly different brain functions.

Perhaps there is a way of arguing that vastly different strategies should have resulted in different networks whereas the updated Figure 2 shows substantial overlap for the two conditions. This makes me wonder whether it is possible to include a control analysis that first extracts voxels in a manner of a conjunction analysis, then conducts the graph-based analysis (i.e., for both conditions together) before back-projecting the results onto the two conditions. But for now, I am not convinced that my concern about a confound is removed.

Response: In both conditions, participants were required to maintain both color and location throughout the delay; only in the binding condition did they additionally need to maintain their conjunction. We agree that participants may adopt different strategies to encode color or location. However, as the reviewer noted, “qualitatively different functions might be involved in the binding condition because it prevents global strategies.” Precisely—that function is feature binding. Thus, the only difference between conditions is whether feature binding occurs.

Moreover, it should be emphasized that our paradigm has been universally adopted in the binding literature across behavioral (Treisman & Zhang, 2006; Wheeler & Treisman, 2002), neuroimaging (Libby et al., 2014; Parra et al., 2014; Piekema et al., 2006; Sala & Courtney, 2007), electrophysiological (Birba et al., 2017; Morgan et al., 2011; Song & Jiang, 2006; Tseng et al., 2016), and clinical domains (Holcomb et al., 2020; Mitchell et al., 2000; Parra et al., 2010; Piekema et al., 2006). Entire

computational models of binding have been constructed using this paradigm (Schneegans & Bays, 2017). To question whether this design measures binding is effectively to disregard the methodological foundation of an entire subfield.

We have added this clarification to the Discussion following the section on task difficulty (see page 12 and below).

“Notably, our study builds on the change-detection paradigm, which has been extensively validated for investigating feature binding in the working memory field. For more than two decades, this paradigm has served as the foundation for hundreds of studies across behavioral (Treisman & Zhang, 2006; Wheeler & Treisman, 2002), neuroimaging (Libby et al., 2014; Parra et al., 2014; Piekema et al., 2006; Sala & Courtney, 2007), electrophysiological (Birba et al., 2017; Morgan et al., 2011; Song & Jiang, 2006; Tseng et al., 2016), and clinical domains (Holcomb et al., 2020; Mitchell et al., 2000; Parra et al., 2010; Piekema et al., 2006). Entire computational models of binding have also been developed based on this paradigm (Schneegans & Bays, 2017). Potential concerns regarding task difficulty or the use of alternative strategies have already been thoroughly addressed and excluded in this body of work.”

Concern 2:

Thanks for the additional clarifications and the extra figure. It is good to hear that a different approach yields similar results thereby demonstrating that the findings are robust. I might have missed it but I didn't find the additional analysis in the manuscript. If it is not included yet then perhaps it could be added to the supplementary material.

Response: Thanks, fixed, with the related description provided on page 12 and figure S8.

Concern 3:

The authors have clearly put significant thought into addressing my third concern. The authors point out that both their experimental conditions show strong involvement of the somatomotor area. However, there are alternative explanations for this activation to do with salience or attention, rather than the formation of visual feature conjunctions. The main difficulty, however, is that there is no differential activation for the two experimental conditions. The cited papers by Frost & Metin (1985), Sieben et al. (2013), Sun et al. (2016), and Taylor-Clarke et al. (2002) show an involvement of the somatomotor area, but their tasks always involve both visual and somatosensory processes. Smith & Goodale (2015) used a purely visual task and found that object classes could be decoded from S1 and S2. This likely reflects top-down projections from high level ventral and/or dorsal areas, or from prefrontal cortex, consistent with the reciprocal connections between the somatomotor area and higher-order regions (Morecraft et al., 2012; Rolls et al., 2023; Shipp et al., 1998). However, such top-down projections don't amount to evidence for binding. In sum, the present findings might involve attentional or saliency functions, they might also involve classification signals reverberating from visual areas but they do not demonstrate conjunctive coding. To demonstrate conjunctive representations, it would

be important to demonstrate that conjunctive information can be decoded from the somatomotor area (e.g., Pollmann et al, NeuroImage, 2014).

Response: It should be clarified that our claim concerns the somatomotor area showing the strongest connectivity with other brain regions, rather than increased local activity (as reflected by BOLD signal differences). This points to a dynamic, network-level process rather than static regional activity. This interpretation is supported by our conditional difference analysis of BOLD signals and the observed enhancement of connectivity patterns within a distributed network during binding. We have revised wording throughout the manuscript to avoid potential misunderstandings and added further discussion on page 14 (see below).

“Notably, the somatomotor area showed the strongest connectivity with other brain regions, rather than increased local activity (as reflected by BOLD signal differences). This suggests a dynamic, network-level role rather than static neural activity. Its specific functional contribution to feature binding, however, remains to be determined and requires further investigation.”

Reviewer #4 (Remarks to the Author):

The task reported here is based on Experiment 3a of Wheeler and Treisman, (2002), and quite a few other studies have also used this sort of whole array either/both color and location change detection task to tease apart binding from maintenance of multiple features. I also agree with the authors that the either/both task used here addresses limitations of the single (remember color or location) vs double (remember both) tasks that have been used more commonly in neuroimaging studies of feature binding. That said, behavioral work in the binding literature (including Wheeler and Treisman) typically relies on multiple variations of the change detection task (i.e., single vs whole display probes, remember one vs remember both features) to isolate which cognitive mechanisms drive performance. So, I also agree with the reviewers that there are limitations of the present design that need to be addressed.

I don't think it's sufficient for the manuscript to simply write-off all the careful nuance of the binding literature with the paragraph at lines 319 - 326. No one is arguing that change detection tasks generally are the problem. Instead, the problem is that the present study ran one specific version of a change detection task, and there doesn't appear to be a consensus in the binding literature that the either/both whole array task used here is the best way to measure feature binding. Indeed, most of the studies cited in the rebuttal and in the paragraph included from 319 - 326 used a different version of the task than the version used in the reported experiment. While it'd be most compelling to see these results replicate with a different version of a binding change detection task, that's probably not a fair request. However, a detailed discussion of the pros and cons of this task paradigm and how the neural findings may or may not change under different task conditions (e.g. a single probe report task) would go a long way in improving the manuscript.

Response: We appreciate your suggestion and have deleted the original one (lines 319 - 326) and substantially expanded the Discussion to articulate the related rationale and to acknowledge its limitations.

We completely agree that behavioral work on feature binding typically employs multiple task variants to isolate distinct cognitive components. Our choice to focus on a single, canonical paradigm was deliberate and theoretically motivated rather than an omission. First, converging evidence shows that binding benefits do not depend on probe format (single-item vs whole-array), but instead on whether spatial location remains diagnostic at test (Kovacs & Harris, 2019; Pertzov & Husain, 2014; Schneegans & Bays, 2017; Wang et al., 2016). Binding costs relative to single-feature conditions are observed under both single-probe and whole-array designs (Wheeler & Treisman, 2002; Treisman & Zhang, 2006). Therefore, the cognitive distinction between binding and either-memory does not hinge on probe format but on the representational requirement (maintain integrated objects vs independent features).

Second, we selected this paradigm specifically because it (a) preserves the functional relevance of location cues, and (b) places bindings under multi-item interference, which is theoretically useful for testing network-level coordination in

fMRI.

That said, we agree that our design cannot isolate all components that multi-variant behavioral paradigms can. We have added a detailed paragraph explicitly discussing these strengths and limitations (see revised manuscript, p. 12):

“Importantly, the present study utilizes a task variant that preserves the diagnostic role of spatial location while imposing multi-item interference at probe (Wang et al., 2016; Wheeler & Treisman, 2002). This design permits a stringent test of conjunction maintenance under matched memory load. Nonetheless, alternative task variants, such as single-item probes, may differ in the degree of probe interference or retrieval demands (Treisman & Zhang, 2006). Our findings should therefore be interpreted as reflecting network-level coordination within this specific task context, rather than a general signature of binding across paradigms. Future work directly comparing binding variants within the same imaging framework will be essential for determining which neural effects generalize across tasks and which remain task-dependent.”

This addition addresses your suggestion for a more transparent treatment of paradigm diversity and interpretive boundaries.